# Marker-free co-selection for successive rounds of prime editing in human cells

Sébastien Levesque[1,2], Diana Mayorga [1,2], Jean-Philippe Fiset[1,2], Claudia Goupil[1,2], Alexis Duringer [1,2], Andréanne Loiselle[1,2], Eva Bouchard[1,2], Daniel Agudelo[1,2] & Yannick Doyon [1,2] ✉

Prime editing enables the introduction of precise point mutations, small insertions, or short deletions without requiring donor DNA templates. However, efficiency remains a key challenge in a broad range of human cell types. In this work, we design a robust co-selection strategy through coediting of the ubiquitous and essential sodium/potassium pump (Na$^+$/K$^+$ ATPase). We readily engineer highly modified pools of cells and clones with homozygous modifications for functional studies with minimal pegRNA optimization. This process reveals that nicking the non-edited strand stimulates multiallelic editing but often generates tandem duplications and large deletions at the target site, an outcome dictated by the relative orientation of the protospacer adjacent motifs. Our approach streamlines the production of cell lines with multiple genetic modifications to create cellular models for biological research and lays the foundation for the development of cell-type specific co-selection strategies.

Prime editing (PE) is a genre of genome editing that can be used to install nucleotide substitutions, as well as, short insertions and deletions without requiring donor DNA or double-strand breaks (DSBs)[1–3]. Prime editors are ribonucleoproteins (RNPs) composed of a Cas9 nickase fused to a reverse transcriptase (RT) and a programmable pegRNAs. The pegRNAs are single-guide RNAs (sgRNAs)[4] extended at their 3' ends with two motifs required for initiation of reverse transcription and specification of the genomic changes, namely the primer binding site (PBS) and the RT template[1]. As compared to sgRNAs for CRISPR nucleases and base editors, the design of pegRNAs is more complex since few PBS and RT template combinations are functional in a broad range of cell lines[1]. While the prediction of efficiency of pegRNAs in human cells is making progress[5,6], there remains a need to test many permutations empirically to find optimal reagents.

To date, six types of prime editors are used: PE2, PE3, PE3b, PE4, PE5 and PE5b[1,7]. PE2 relies on an engineered Moloney murine leukemia virus (M-MLV) reverse transcriptase (RT) fused to a SpCas9 nickase (H840A) to introduce small genetic changes templated by the pegRNA. After DNA binding and nicking, PE2 reverse transcribes the intended edit from the pegRNA 3' extension. The resolution of the heteroduplex, by a mechanism that remains incompletely defined, results in the introduction of the intended edit[1–3,7]. The PE3 strategy relies on an additional sgRNA to nick the non-edited strand and direct DNA repair to preferentially incorporate the intended edit which typically results in higher editing efficiency versus PE2[1,3]. However, PE3 can yield a low but detectable rate of indels formation at the target site, decreasing product purity[1,3,9]. When permitting, a nicking sgRNA that matches the edited strand can be used to minimize the presence of concurrent nicks, and consequently, indel formation[1]. Denoted as PE3b[1], this strategy is not applicable when no protospacer adjacent motif (PAM) is available near the targeted edit. Transient coexpression of a dominant negative DNA mismatch repair (MMR) protein (MLH1dn) along with PE components further enhances efficiency and lowers editing byproducts[7]. This strategy yielded the PE4 (PE2 + MLH1dn), PE5 (PE3 + MLH1dn), and PE5b (PE3b + MLH1dn) editors[7]. In addition, nuclease prime editors have been shown to improve PE efficiency, but it comes at the expense of product purity[10–12]. Thus, the optimal PE strategy to adopt varies according to the context and there is a need to develop methods to consistently improve its success rate.

[1]Centre Hospitalier Universitaire de Québec Research Center—Université Laval, Québec, QC G1V 4G2, Canada. [2]Université Laval Cancer Research Centre, Québec, QC G1V 0A6, Canada. ✉e-mail: Yannick.Doyon@crchudequebec.ulaval.ca

To circumvent these limitations several approaches have been implemented including editing with purified ribonucleoprotein complexes (RNPs)[13], mRNA-based delivery[7,14], engineered pegRNAs (epegRNAs)[15], NLS- and codon-optimized prime editors[7,9,16], enrichment with puromycin, and fluorescent reporter-based selection[10,17,18]. Nevertheless, these improvements are still dependent on extensive pegRNA optimization, absolute activity remains low, and the streamlined production of homozygous cell lines has yet to be achieved[1,15]. Hence, further refinements to these approaches are needed to boost the efficiency of PE in human cells[2,3,7,15].

We previously reported a robust co-selection method for CRISPR nucleases that relies on co-targeting a gene of interest (GOI) and the *ATP1A1* locus to confer dominant cellular resistance to ouabain[19], a plant-derived inhibitor of the ubiquitous and essential Na$^+$/K$^+$ ATPase[20,21] (Fig. 1a and Supplementary Fig. 1). Such an approach derives from the observation that simultaneous targeting of two different loci results in double-editing events that are not statistically independent[19,22–27]. The strategy has proven to be portable to non-homologous end joining (NHEJ), homology directed repair (HDR), and base editing events in near-haploid, diploid

and polyploid cells[19,28–34]. To further extend the versatility of this method, we took advantage of the extensive mutagenesis and crystallographic studies that have led to the identification of mutated versions of the enzyme that are resistant to various concentration of ouabain[20,21,35–37]. We engineered these dominant gain-of-function mutations in *ATP1A1* to confer cellular resistance to discrete levels of ouabain, allowing sequential rounds of selection to be performed, by progressively increasing the concentration of the inhibitor at each step. This approach was adapted to enrich PE events in human cells whilst enabling multiple rounds of co-selection. Lastly, we observed that product purity is decreased when using a secondary nick to stimulate PE and that these events are often missed when sequencing target sites in bulk populations of cells. Hence, we propose an easy-to-implement toolkit to select cells engineered via PE without the use of exogenous selection markers that largely bypasses the need for extensive pegRNA optimization. This approach is compatible with recent improvements of the PE technology and should simplify the generation of isogenic panels of human cell lines with multiple genetic modifications.

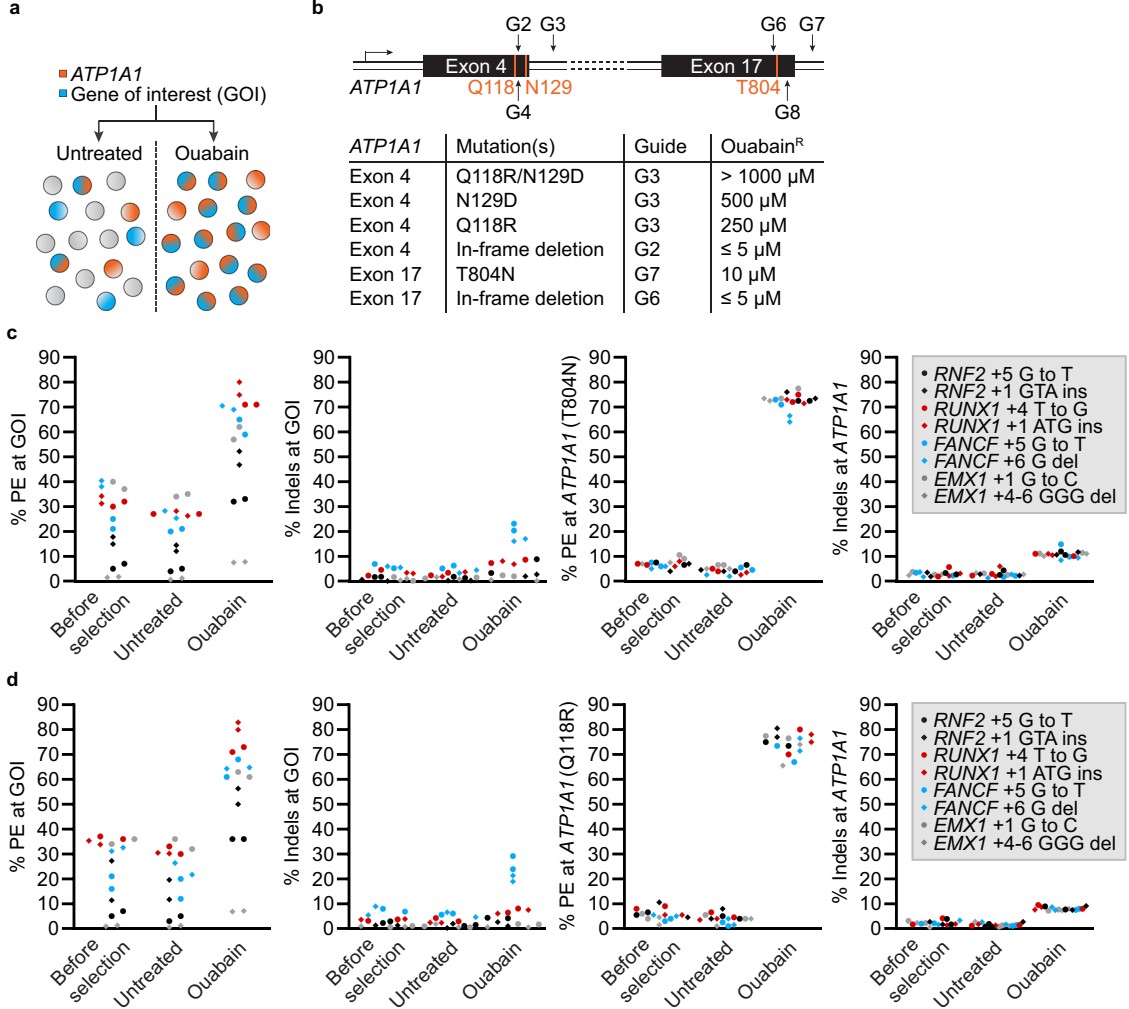

**Fig. 1 | Robust co-selection for prime editing. a** Schematic for the co-enrichment of CRISPR-driven editing events at a GOI. **b** Schematic representation of the *ATP1A1* locus regions targeted by SpCas9. The first and third extracellular loops of the Na$^+$/K$^+$ ATPase are encoded by *ATP1A1* exon 4 and 17, respectively. The relative levels of resistance to ouabain conferred by different *ATP1A1* mutations in K562 cells are shown. **c** PE and small indels quantification as determined by BEAT and TIDE analysis from Sanger sequencing. K562 cells were transfected with PE3 vectors

targeting *ATP1A1* exon 17 (T804N) and the indicated GOI. Genomic DNA was harvested 3 days post-transfection (before selection) and cells were treated (ouabain) or not (untreated) with 0.5 μM ouabain until all non-resistant cells were eliminated. **d** Same as in **c** but co-targeting was performed via *ATP1A1* exon 4 (Q118R). $n = 2$ independent biological replicates performed at different times. Source data are provided in the Source Data file.

## Results

### *ATP1A1* variants conferring discrete levels of resistance to ouabain

The main subunit of the Na$^+$/K$^+$-ATPase (a.k.a sodium-potassium pump), encoded by *ATP1A1*, is a 1023 amino acid long protein containing ten transmembrane domains and five extracellular loops which transports Na$^+$ and K$^+$ against their electrochemical gradients. It is also the extracellular receptor for cardiac glycosides, a class of drugs that comprises ouabain and digoxin[20,21,38]. A long history of random mutagenesis studies has mapped amino acid residues that alter ouabain sensitivity[36,39]. We previously demonstrated that creating in-frame deletions within *ATP1A1* exon 4, which encodes the first extracellular loop, conferred resistance to low doses (<5 μM) of ouabain while point mutations (Q118R and N129D) enabled K562 cells to grow in the presence of ~1000 μM ouabain (Fig. 1b and Supplementary Fig. 1). To further increase the diversity of variants for co-selections, we tested two additional mutations (L800K and T804N) located in the third extracellular loop[36,37,40,41].

We designed a sgRNA to cleave within *ATP1A1* intron 17 (hereafter named G7) in the vicinity of T804 and used a single-stranded oligodeoxynucleotide (ssODN) to introduce either L800K or T804N (Supplementary Fig. 1). Cleaving within the intron allows for seamless induction of recombination within the juxtaposed exon and mitigates the negative impact caused by on-target NHEJ-based mutagenesis of this essential gene[19]. K562 cells were transfected with sgRNA G7 along with ssODNs, and growth was monitored after the addition of a low dose of ouabain (0.5 μM) which is sufficient to kill all non-edited cells within 48 h[19]. Only cells bearing the T804N mutation survived selection and titration of ouabain revealed that growth remained robust up to 10 μM ouabain (Fig. 1b). In addition, cleavage within exon 17 selects for in-frame deletions changing the extracellular loops of the pump and conferring resistance to low concentrations of ouabain (Fig. 1b and Supplementary Fig. 1). Thus, dominant gain-of-function mutations can also be generated by editing the third extracellular loop of the Na$^+$/K$^+$-ATPase and confer an intermediate level of resistance to the drug.

We hypothesized that these mutations could be engineered in succession to create variants of the pump that are progressively more resistant to the selection agent. By targeting *ATP1A1* repeatedly, and increasing the dose of ouabain at each step, one could perform sequential rounds of co-selection and engineer cell lines with multiple genetic modifications. As a starting point, we used a homogenous population of cells expressing mScarlet-I from a cassette nested within *ATP1A1* intron 17[42]. This pool was created by targeted integration and selection with 0.5 μM ouabain via the co-introduction of T804N (Supplementary Fig. 2). In this cell line, we tagged *NPM1* at its N-terminus by targeted insertion of the coding sequence for mNeonGreen (mNG). Following co-selection using 100 μM ouabain via Q118R/N129D, 57% of cells expressed the mNG-NPM1 fusion based on FACS analysis. Targeted integration was further confirmed by out-out PCR (Supplementary Fig. 3). Similarly, it was possible to target an expression cassette for mNG within *ATP1A1* intron 4 creating cell lines that expressed both transgenes in 93% of the cells upon selection with 100 μM ouabain (Supplementary Fig. 3). Therefore, this pool of selected cells had two nested and functional expression cassettes within introns of *ATP1A1* as confirmed via out-out PCR (Supplementary Figs. 2 and 3). mScarlet-I expression was maintained in all cells demonstrating sustained function over two rounds of selection (Supplementary Fig. 3). Finally, we were able to perform three sequential steps by starting with NHEJ-mediated editing at *ATP1A1* which confers the lowest level of resistance to ouabain (Supplementary Fig. 4). Taken together, these results demonstrate that *ATP1A1* can be targeted multiple times to perform sequential rounds of marker-free selection by increasing the dose of ouabain at each step. Overall, this efficient and versatile system simplifies the engineering of human cells containing multiple genetic modifications.

### Robust co-selection for prime editing events

Despite continuous improvements in PE technology, there remains a need to develop universal methods to achieve high levels of editing in bulk populations, especially in MMR-proficient cells[1,7,10,43]. As observed with CRISPR-nucleases and base editors[19,28] (see also Supplementary Fig. 5), we hypothesized that cells that are proficient at completing prime editing at the *ATP1A1* locus are more likely to harbor a second prime editing event at a GOI. We first designed pegRNAs to install the T804N and Q118R mutations and confirmed that PE3 conferred resistance to ouabain without affecting proliferation of K562 cells (Supplementary Figs. 6 and 7). In addition, steady-state levels of ATP1A1 were not affected by the editing process as determined by western blotting (Supplementary Fig. 7). While the PE2 strategy allowed the isolation of ouabain-resistant single-cell derived clones harboring one prime edited allele (46/47 monoallelic clones, see Supplementary Table 1), detectable levels of editing in bulk populations of cells were only observed when using a nick sgRNA (PE3) so the later approach was selected for co-selection.

To facilitate the process, we constructed tandem U6-driven pegRNA and nick-inducing sgRNA expression vectors to co-target *ATP1A1* and GOIs using a three vectors PE3 system (Supplementary Fig. 8). We selected eight previously optimized pegRNAs[1] targeting four loci to introduce point mutations, insertions, and deletions to test the impact of co-selection in K562 and HeLa S3 cells. These MMR-proficient cell lines are known to display lower PE rates, as opposed to HEK293 cells, and have been extensively used as a model system for PE[1,7,10]. K562 cells were transfected with PE3 elements and expanded in the presence or absence of 0.5 μM ouabain starting at D3 post-transfection until all non-resistant cells were eliminated, which typically takes 10 days. The frequencies of alleles harboring precise modifications markedly increased after co-selection for every pegRNA tested, as determined by BEAT[44] and TIDE[45] analysis (Fig. 1c, d). Co-targeting with either PE3-T804N or PE3-Q118R yielded very similar improvements in editing rates without exacerbating indels byproducts[1] (Fig. 1c, d). While we could reproducibly detect diverse translocations between chromosome 1 (*ATP1A1* exons 4 or 17) and chromosome 2 (*EMX1*) and 21 (*RUNX1*) when using wild-type SpCas9, it was not the case with PE3 at 3 days post-transfection (Supplementary Fig. 9). However, we could amplify rare translocation events in PE3 samples from selected pools. Direct sequencing of the amplicons further confirmed the rarity of PE3-induced translocations (Supplementary Fig. 9). Comparable trends were observed in HeLa S3 cells despite slightly lower absolute levels of efficacy with the most active pegRNAs reaching 59% editing, as compared to 83% in K562 cells (Supplementary Fig. 8). In HeLa S3, some pegRNAs with undetectable activity prior to selection reached between 26% and 31% upon co-selection (Supplementary Fig. 8). Similar improvements were observed in an EBFP to EGFP reporter system in PE2, PE3, and PE3b modes (Supplementary Fig. 10). Thus, co-selection improves yields for all types of CRISPR-based editing systems.

### Successive rounds of co-selection for prime editing in human cells

We tested if cells having undergone a first round of co-selection at GOI A with PE3-T804N were competent for a second round of co-selection at GOI B with PE3-Q118R (Fig. 2a). A pool of K562 cells with 80% of alleles harboring the *RUNX1* + 1 ATG insertion (GOI A) pre-selected using 0.5 μM ouabain was used as starting material (see Fig. 1c). This cell pool was independently transfected with six different pegRNAs (GOIs B) before a second round of co-selection was performed using 100 μM ouabain. This two-step process yielded highly modified cell pools at the two GOIs (Fig. 2b). Similar results were obtained when the

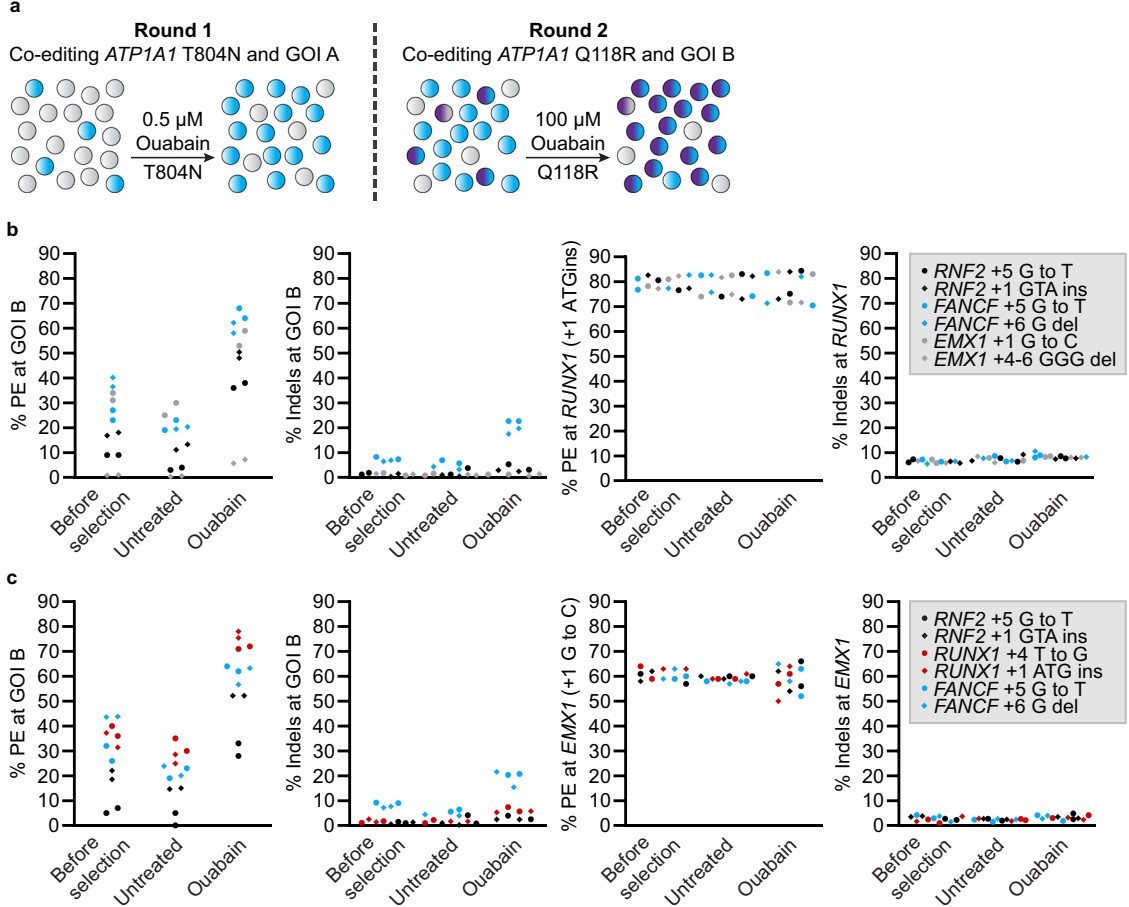

**Fig. 2 | Robust co-selection for successive rounds of prime editing. a** Schematic of the strategy for performing successive rounds of co-selection. Cells harboring modifications at *ATP1A1* (T804N) and GOI A are first co-selected with 0.5 μM ouabain. Following the first round, a subsequent round of co-selection occurs at GOI B via modification at *ATP1A1* (Q118R) with 100 μM ouabain. **b** PE and small indels quantification as determined by BEAT and TIDE analysis from Sanger sequencing. K562 cells harboring the *ATP1A1*-T804N and *RUNX1* +1 ATG insertion (GOI A)

modifications (Fig. 1c) were transfected with PE3 vectors targeting *ATP1A1* exon 4 (Q118R) and the indicated GOI B. Genomic DNA was harvested 3 days post-transfection (before selection) and cells were treated (ouabain) or not (untreated) with 100 μM ouabain until all non-resistant cells were eliminated. **c** Same as in **b** with K562 cells harboring the *ATP1A1*-T804N and *EMX1* +1 G to C (GOI A) modifications (See Fig. 1c). *n* = 2 independent biological replicates performed at different times. Source data are provided in the Source Data file.

entire procedure was reproduced using a pool of cells with the *EMX1* +1G to C modification as starting material (Fig. 2c). For example, one pool contained 64% and 81% edited alleles at *FANCF* and *RUNX1*, respectively. Another pool had 78% and 63% editing at *RUNX1* and *EMX1*, respectively. The level of enrichment in these subsequent rounds of PE paralleled the improvements observed in the first rounds of editing, demonstrating that the two steps of co-selection perform equally well and are independent (compare Figs. 1 and 2). As observed for the single *ATP1A1* mutations, selected bulk populations of cells harboring both T804N and Q118R mutations grew robustly and displayed little to no decrease in ATP1A1 expression (Supplementary Fig. 7). These data demonstrate that two rounds of PE can be performed sequentially to considerably enrich cells stably modified at two positions within the genome.

### Installation of clinically relevant mutations at *MTOR*

The serine/threonine kinase mTOR is a master regulator of eukaryotic cell growth and metabolism that integrates environmental cues. Given its central role in maintaining physiological homeostasis, mTOR mutations lead to a diverse range of diseases, including cancers[46–49]. To determine if co-selection could streamline the production of homozygous cell lines for functional studies, we designed pegRNAs to install well-characterized and clinically relevant *MTOR* hyperactivating mutations[47,48,50,51] or rapamycin resistance mutations[52–54]. While the

design and screening of several pegRNAs remains a critical aspect of successful PE[1,7,15], we tested only one pegRNA per target based on previously established rules[1,5] for this proof-of-concept experiment. We designed pegRNAs harboring 25 nucleotides (nts) 3' extensions with 12- to 13-nts PBS and PAM or seed mutations within the RT template to install four hyperactivating mutations (L2431P, E2419K, A2020V, I2017T) and two mutations causing resistance to rapamycin (F2108L, S2035T). To monitor the impact of mTOR variants on signaling activity, PE was performed in a K562 cell line containing an adapted mTOR signaling indicator (mSc-TOSI)[55] targeted to *ATP1A1* intron 17 (Fig. 3a and Supplementary Fig. 11). In this system, active mTORC1 signaling results in rapid phosphorylation of the mSc-TOSI phosphodegron by S6K, ubiquitination, and degradation by the proteasome while its inhibition stabilizes the reporter (Supplementary Fig. 11)[55]. The mTOR reporter cell line was transfected with the indicated pegRNAs in a PE3 format and cells were treated or not with 100 μM ouabain starting 3 days post-transfection. At D14, genomic DNA was extracted and the percentage of PE alleles or indels was quantified. In the selected populations, E2419K increased 4 fold (from 21 to 87%), F2108L went up 5 fold (from 12 to 58%), and I2017T rose 10 fold (from 4 to 39%) (Fig. 3b). While undetectable before selection, co-selection allowed the detection of 12%, 9%, and 6% of alleles harboring L2431P, S2035T, and A2020V, respectively (Fig. 3b). Small indels remained low with (≤8%) after co-selection with all *MTOR* pegRNAs

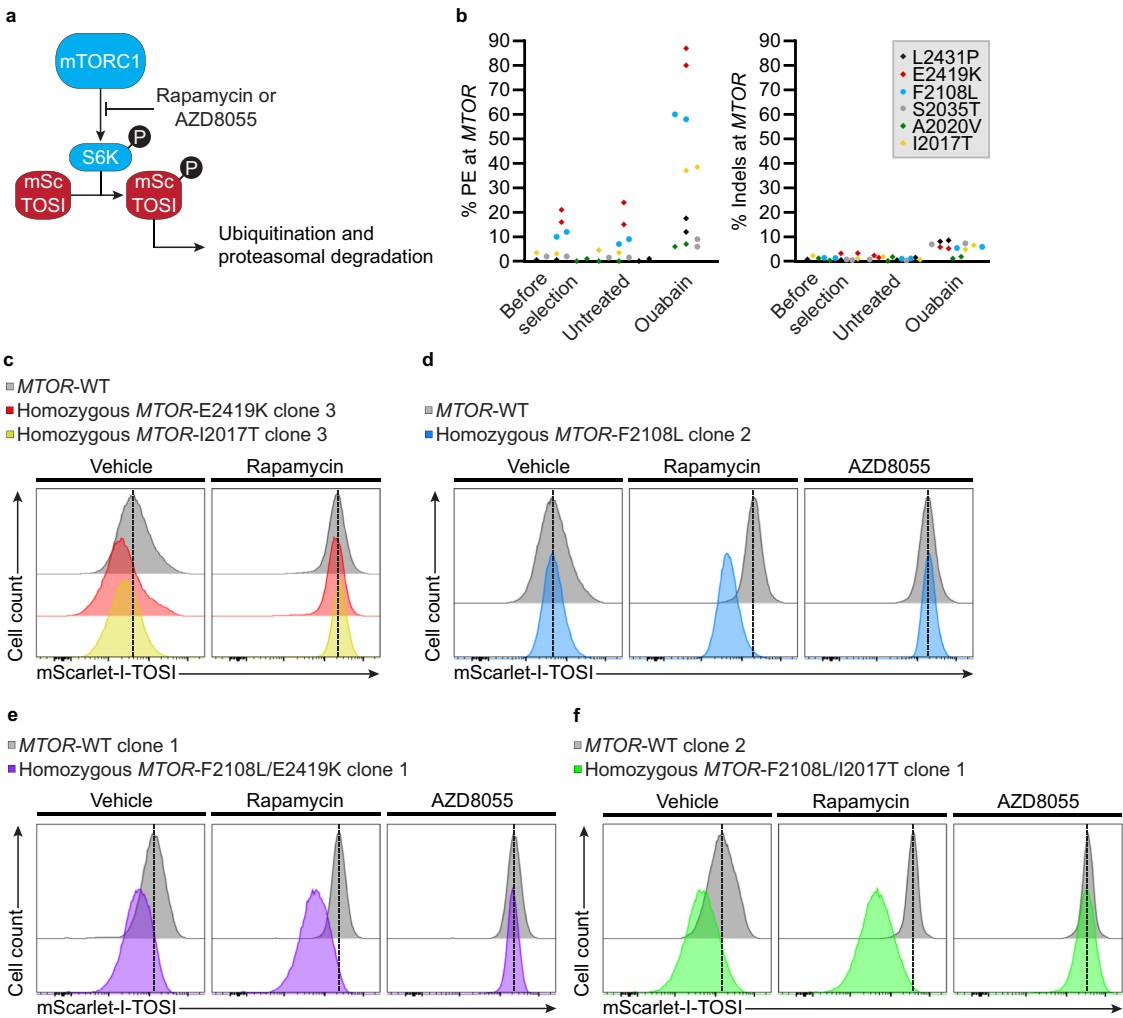

**Fig. 3 | Co-selection for the installation of clinically relevant mutations at the *MTOR* locus. a** Schematic of mScarlet-I mTOR Signaling Indicator (mSc-TOSI) reporter degradation under mTORC1 signaling. **b** PE and small indels quantification as determined by BEAT and TIDE analysis from Sanger sequencing. K562 cells stably expressing the mSc-TOSI reporter were transfected with PE3 vectors targeting *ATP1A1* exon 4 (Q118R) and *MTOR*. Genomic DNA was harvested 3 days post-transfection (before selection) and cells were treated (ouabain) or not (untreated) with 100 μM ouabain until all non-resistant cells were eliminated. *n* = 2 independent biological replicates performed at different times. **c** Histogram plot of mSc-TOSI intensity in homozygous single cell-derived K562 clones harboring *MTOR* hyperactivating mutations. **d** Same as in **c** with a homozygous clone harboring the *MTOR*-F2108L rapamycin resistance mutation. **e** Same as in **c** with a homozygous clone harboring the *MTOR*-F2108L rapamycin resistance mutation and the *MTOR*-E2419K hyperactivating mutation. **f** Same as in **c** with a homozygous clone harboring the *MTOR*-F2108L rapamycin resistance mutation and the *MTOR*-I2017T hyperactivating mutation. Where indicated, cells were treated for 24 h with 50 nM rapamycin or 50 nM AZD8055 before FACS analysis. Representative images are from one of two independent biological replicates performed at different times with equivalent results (see Supplementary Figs. 15 and 16). Source data are provided in the Source Data file.

(Fig. 3b). Notably, hyperactive mTOR signaling could be detected in co-selected cell pools indicating that the process provides the means to evaluate variant causality in populations of PE edited cells (Supplementary Fig. 12). Similarly, rapamycin resistance conferred by the F2108L mutation could also be observed in the co-selected population (Supplementary Fig. 12).

To test whether co-selection can facilitate the isolation of homozygous clones, we isolated and characterized single cell-derived K562 clones with E2419K, F2108L, and I2017T mutations at *MTOR*. At D3 post transfection, cells were either selected in pools or plated in methylcellulose-containing media with 100 μM ouabain to derive clones. This way, PE efficiency can be compared between pools (see Fig. 3b) and clones. Genomic DNA was extracted from each clone and the region surrounding the targets were amplified by PCR. Amplicons were analysed on agarose gels and by Sanger sequencing coupled to trace decomposition by BEAT[44], DECODR[56], and TIDE[45] (Supplementary Figs. 13 and 14). While K562s contain three copies[57] of both *MTOR* and *ATP1A1*, 34% (30/88) of the clones were homozygous for the desired

*MTOR* mutations. Moreover, biallelic and monoallelic modifications were observed in 25% (22/88) and 34% (30/88) of the clones, respectively. Taking all three *MTOR* mutations together, only 5 clones did not have at least one pegRNA-specified allele out of 88 clones analysed (Table 1 and Supplementary Fig. 14). Large insertions at the target sites were observed when running PCR products on agarose gels (see below). Overall, the many genotypes observed at the clone level indicate that polyclonal populations are created during co-selection.

The functional impact of PE-mediated *MTOR* mutations was tested in single cell-derived clones. As expected, degradation of the mSc-TOSI reporter was intensified in homozygous clones harboring E2419K or I2017T hyperactivating mutations (Fig. 3c). In contrast, monoallelic E2419K and I2017T clones containing inactivating indels at the remaining alleles displayed crippled mTOR signaling despite growing indistinguishably from all other clones (Supplementary Fig. 15). This observation suggests that a single edited allele is not sufficient to hyperactivate mTORC1 signaling in K562 cells, highlighting the importance of carefully genotyping PE-generated clones. Complete

**Table 1 | Distribution of alleles in single cell-derived clones edited at *MTOR* by co-selection**

| *MTOR* mutation | % of clones with indicated alleles | | | | |
|---|---|---|---|---|---|
| | WT | WT + large insertion(s) | Monoallelic PE* | Biallelic PE* | Triallelic PE& |
| E2419K | 0% (0/30) | 0% (0/30) | 40% (12/30) | 33% (10/30) | 27% (8/30) |
| F2108L | 0% (0/31) | 6% (2/31) | 26% (8/31) | 19% (6/31) | 48% (15/31) |
| I2017T | 4% (1/27) | 11% (3/27) | 37% (10/27) | 22% (6/27) | 26% (7/27) |
| Total | 1% (1/88) | 6% (5/88) | 34% (30/88) | 25% (22/88) | 34% (30/88) |

Compilation of genotypes from Supplementary Fig. 14.
*Except for two *MTOR*-F2108L clones, all remaining alleles harbor small indels, large insertions and deletions, or pegRNA scaffold incorporation (See Supplementary Fig. 14).
&Homozygous, only PE-specified sequences are detected.

rapamycin resistance was also observed in homozygous F2108L clones (Fig. 3d and Supplementary Fig. 15). First and second generation mTOR inhibitors[54,58] were used to demonstrate the specificity of the F2108L mutations and to show that the reporter still responded to drug treatment after clonal derivation (Fig. 3d and Supplementary Fig. 15). In addition, all clones used for functional FACS-based assays were homozygous for mSc-TOSI knock-in at *ATP1A1* intron 17 as determined by out-out PCR further indicating that the observed variation in reporter signal were caused by the presence of the desired *MTOR* mutations (Supplementary Fig. 15). Note that the wild-type (WT) *MTOR* controls used in these FACS-based assays contained the *ATP1A1* Q118R mutation and were selected with ouabain just like the test samples to mitigate any potential impact of the co-selection process itself on the readout of the experiments[57]. Taken together, these data indicate that homozygous clones can be readily isolated via marker-free co-selection to characterize pathogenic mutations even when low levels of PE are observed in the starting population.

We then performed successive rounds of co-selection to isolate cells harboring both hyperactivating and rapamycin resistance mutations (Supplementary Fig. 16). The successive rounds of co-selection allowed the enrichment of highly modified population of cells. For example, one pool contained 88% E2419K and 58% F2108L alleles, and another pool had 41% I2017T and 54% F2108L alleles (Supplementary Fig. 16). FACS-based analysis in single cell-derived clones revealed complete rapamycin resistance and elevated mTORC1 signaling in homozygous double mutant cell lines (Fig. 3e, f). Double *MTOR* mutants remained sensitive to the second generation kinase inhibitor AZD8055 (Fig. 3e, f). While co-selection greatly facilitated the isolation of homozygous single cell-derived clones, only 2 and 1 out of 16 clones were homozygous for F2108L/E2419K and F2108L/I2017T, respectively. The presence of large insertions at one of the two targeted genomic regions in most clones presented a challenge for efficient isolation of double homozygous mutants (see below).

In HeLa S3 cells, PCR-based genotyping revealed that the percentage of alleles harboring the E2419K, F2108L, and I2017T mutations increased 5 fold for E2419K (from 17 to 78%), 4 fold for F2108L (from 9 to 39%), and 8 fold for I2017T (from 4% to 30%) (Supplementary Fig. 17). The ratio between PE and short indels was lower in HeLa S3 cells than in K562s (Fig. 3b and Supplementary Fig. 17). In the pools of co-selected HeLa S3 cells, larger PE3-driven insertions were observed in gel-based assays which were not detected under the same conditions in K562s (Supplementary Fig. 17 and below). In U2OS cells, we initially failed to detect activity with these pegRNAs. Hence, we combined epegRNAs (tevopreQ1[15]), the PEmax architecture, and MMR-evading mutations to further enhance activity (Supplementary Fig. 18)[7,15]. Consequently, markedly increased prime editing rates and efficient ouabain co-selection were achieved in U2OS for two out of three targets tested reaching 89% for the E2419K mutation and 74% for the F2108L mutation (Supplementary Fig. 19). PE3 and PE5 (PE3 + MLH1dn) approaches yielded similar results in this context (Supplementary Fig. 19). Altogether, co-selection is compatible with new PE

enhancements, allowing highly efficient genome editing in more challenging settings.

## Characterization of PE3 editing byproducts

When genotyping individual K562 clones modified at *MTOR* via PE3, we observed a wide range of relatively short PE-driven indels including tandem duplications and deletions of sequences found between the two sites of nicking[7,13] (Supplementary Fig. 13). This spectrum of byproducts is dissimilar to the -1–5 bp indels typically observed with Cas9 nucleases and has been previously reported[59]. In fact, most clones (55/58) that were not homozygous for the PE3-specified *MTOR* mutations contained undesired insertions, deletions or pegRNA scaffold incorporations (Table 1 and Supplementary Figs. 13, 14). Large (>50 bp) PE-mediated insertions and a few large deletions were also frequently observed via gel-based assays (Supplementary Fig. 14). Similar insertions also occur in HeLa S3 cells during PE3 (Supplementary Fig. 17). These observations contrast with the relatively low levels of small indels detected in cell populations with these pegRNAs (Fig. 3b). This suggests that PCR bias in bulk population of cells may lead to an underrepresentation of larger insertions products driven by PE3[59].

To further characterize these PE3 byproducts, we cloned and sequenced several large insertions from single cell-derived K562 clones edited at *MTOR* (Fig. 4 and Supplementary Fig. 14). Tandem duplications formed by sequences found between the original nicking sites were observed irrespective of the pegRNAs tested. Often present in multiple copies these insertions could reach up to 1 kb (Fig. 4b). We note that most target sites and PAMs were not modified which could have allowed for consecutive rounds of nicking and repair leading to the formation of tandem duplications. However, insertions created by the *MTOR-E2419K* pegRNA also contained the PE3-specified edit. As this mutation changes the canonical NGG to the permissive NGA PAM, it is unclear if the site could still be re-targeted or if PE occurred post amplification (Fig. 4b). A similar DNA repair profile was also observed in bulk populations of HeLa S3 cells, confirming that this type of large PE3-mediated insertions is not limited to K562 cells (Supplementary Fig. 20). Alongside duplications, DNA sequences that did not share homology with the targeted locus were also found. For example, a large 216 bp sequence that shared 100% identity with the amplified *BCR* gene was trapped at the target site (Fig. 4b). We also found two large repeated sequences from the U6-pegRNA vector integrated at the *MTOR* locus in HeLa S3 cells (Supplementary Fig. 20).

These editing byproducts are reminiscent to the DSB repair response resulting from presence of concomitant nicks[59]. Mechanistically, fill-in synthesis of the 3' protruding ends generated by concomitant nicks via Polα-primase could account for these tandem duplications in a pathway that requires 53BP1[60–62]. We thus tested whether the inhibition of 53BP1 with i53[63] could prevent the formation of large PE3-mediated insertions by delaying rapid and predominant NHEJ-mediated repair. However, we did not observe a constant and profound decrease in the presence of large insertions or in the frequency of small indels in bulk populations of HeLa S3 and K562 cells

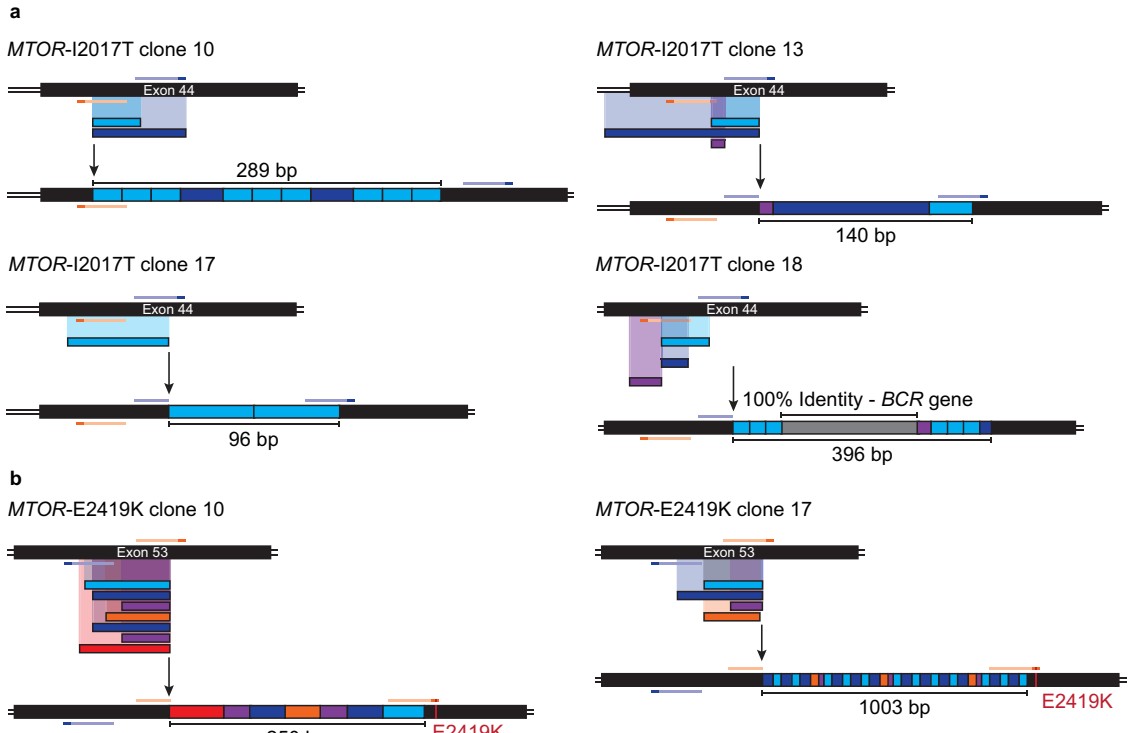

**Fig. 4 | Identification and characterization of large PE3-mediated insertions in single cell-derived clones. a** Schematic representation of the large insertions observed at *MTOR* exon 44 from single cell-derived K562 clones harboring the *MTOR*-I2017T hyperactivating mutation. K562 cells stably expressing the mSc-TOSI reporter were transfected with PE3 vectors targeting *ATP1A1* exon 4 (Q118R) and *MTOR*. Single cell-derived clones were isolated in methylcellulose-based semi-solid RPMI media supplemented with 100 μM ouabain and genomic DNA was harvested after co-selection. TOPO cloning and Sanger sequencing were performed to characterize the large insertions. The pegRNA and its PAM sequence are represented in orange and dark orange, respectively. The nick sgRNA and its PAM sequence are represented in blue and dark blue, respectively. **b** Same as in **a** for large insertions at *MTOR* exon 53 from single cell-derived K562 clones harboring the *MTOR*-E2419K hyperactivating mutation.

(Supplementary Fig. 21). The same was true when overexpressing a dominant negative MLH1 to inhibit MMR as described in the PE5 strategy[7] (Supplementary Fig. 21). Overall, our observations suggest that large PE3-mediated insertions are frequent and underestimated when using standard genotyping methods.

## Secondary nick location dictates the type of PE3 editing byproducts

Considering that the polarity of the overhang structure released after paired Cas9 nicking is a critical determinant of double-strand break repair[59], we hypothesized that the relative positions of the pegRNA nick and complementary-strand nick sites might impact the type of PE3 byproducts. We noticed that our PE3 designs for *MTOR* were all configured with PAMs facing outwards with respect to each other (PAM-out)[59] (Supplementary Fig. 13). Fortunately, at *ATP1A1*, the PE3-Q118R (Exon 4) and PE3-T804N (Exon 17) designs were configured in PAM-out and PAM-in (PAMs facing inwards) configurations, respectively (Fig. 5a). When combined with the H840A SpCas9 nickase (inactivated HNH domain) used in PE3, the PAM-out configuration is predicted to create DSBs with 3' ssDNA protruding ends of 50 bases at *ATP1A1* exon 4 (PE3-Q118R) while the PAM-in configuration at *ATP1A1* exon 17 (PE3-T804N) would create 24 bases 5' ssDNA protrusions (Fig. 5a). Using single cell-derived K562 clones isolated after successive rounds of co-selection, thus bearing both Q118R and T804N mutations, we performed PCR-based genotyping at both sites within *ATP1A1*. This analysis revealed that most large insertions correlate with PAM-out configuration (Fig. 5b, c). We thus tested if converting the PE3-Q118R PAM-out configuration into a PAM-in mode would decrease the frequency of large insertions. We designed three different sgRNAs to nick the complementary-strand at various distances from the pegRNA-specified nick in a PAM-in

arrangement (Fig. 5d). In 51 single cell-derived ouabain resistant clones, the conversion to PAM-in mode abolished large PE3-mediated insertions (Fig. 5e). Taken together these results indicate that PE strategies that lead to the creation of DSBs with 3' overhangs often lead to the formation of tandem duplications at the target site.

To further explore this phenomenon, we setup an EBFP to EGFP reporter system with multiple complementary-strand nick locations (Fig. 6a). We established a K562 cell line homozygous for the targeted integration of an EBFP expression cassette within *ATP1A1*, thus bearing 3 copies of the reporter (Supplementary Fig. 22). In addition, since standard pegRNAs can be degraded by exonucleases, leading to "dead" pegRNA that may only direct nicking by the prime editor, we used epegRNAs containing a structured RNA pseudoknot (tevopreQ1)[15]. In this system, FACS-based analysis reports PE outcomes indirectly; (i) EBFP(+)/EGFP(+) cells result from monoallelic or biallelic PE, (ii) EBFP(−)/EGFP(+) cells are generated by either triallelic PE or combinations of mono- and biallelic PE along with indels, (iii) EBFP(−)/EGFP(−) occur from indels on the three copies of the reporter (Fig. 6b and Supplementary Fig. 22). Indels are defined broadly in this context as any edits that inactivates the reporter. As determined by sequencing pools of co-selected cells, all secondary nick-inducing guides (G1 to G7) were functional. PE3 was more efficient than PE3b, while PE2 was the least active configuration (Supplementary Fig. 22). Detectable levels of small indels were found only in PE3-derived samples (Supplementary Fig. 22). FACS analysis revealed that the vast majority of PE2 outcomes were mono or biallelic PE with a small percentage of cells with modifications of the three alleles (Fig. 6c and Supplementary Fig. 23). PE3b had a similar profile but a higher percentage of editing on the three alleles occurred (Fig. 6c and Supplementary Fig. 23). Of note, complete inactivation of the reporter did not occur with either PE2 or PE3b

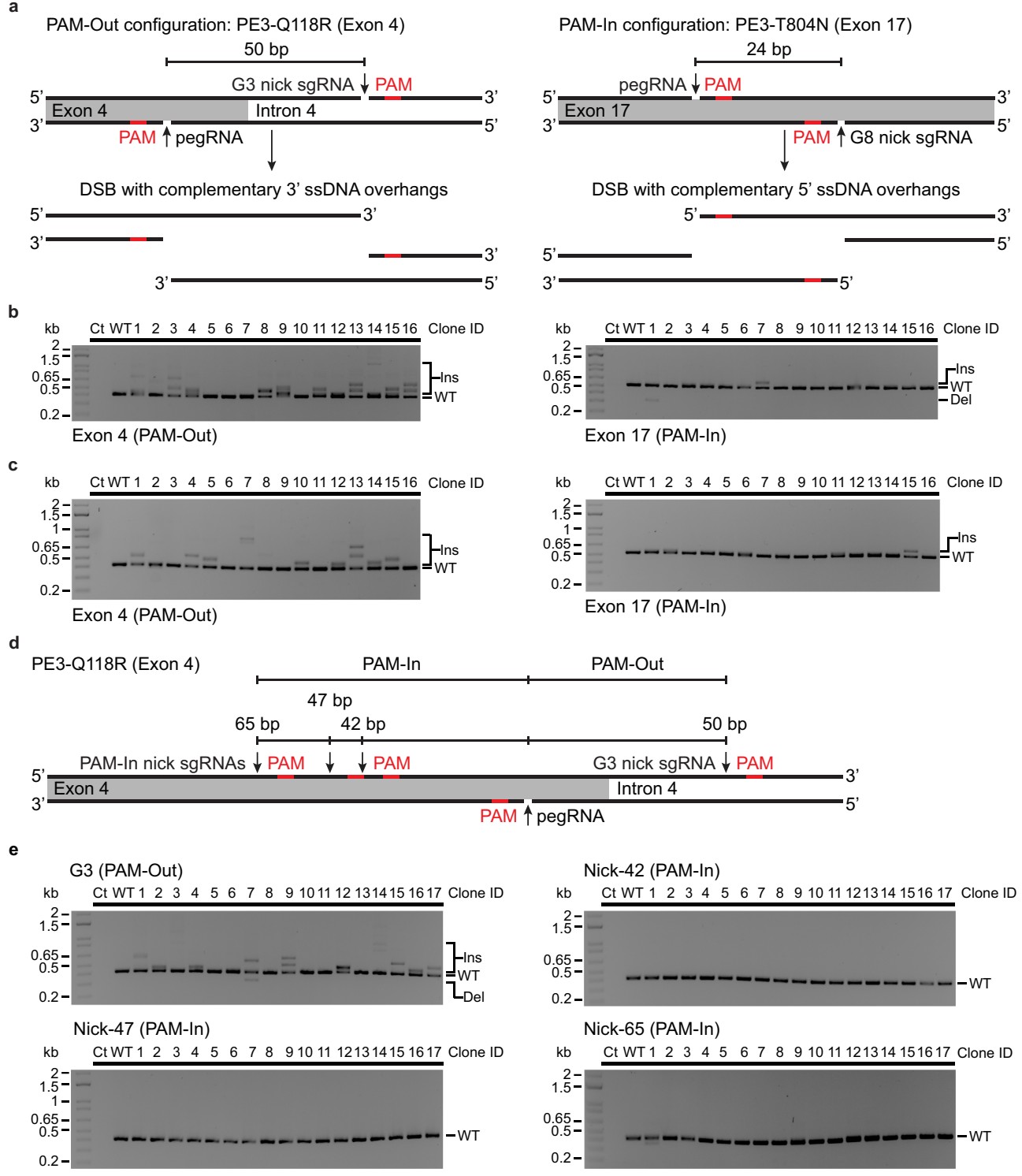

**Fig. 5 | Nick location dictates the type of PE3 editing byproducts at *ATP1A1*.**
**a** Schematic representation of the complementary 5′ and 3′ single-stranded DNA overhangs generated with the PE3-T804N strategy at *ATP1A1* exon 17 (PAM-In configuration) and the PE3-Q118R strategy at *ATP1A1* exon 4 (PAM-Out configuration), respectively. **b** PCR-based genotyping of *ATP1A1* exon 17 and 4 from single cell-derived *MTOR*-F2108L/I2017T K562 clones. Single cell-derived clones were isolated in methylcellulose-based semi-solid RPMI media supplemented with 100 μM ouabain and genomic DNA was harvested after co-selection. *n* = 16 single cell-derived clones from one experiment. **c** Same as in **b** with single cell-derived

*MTOR*-F2108L/E2419K K562 clones. *n* = 16 single cell-derived clones from one experiment. **d** Schematic representation of pegRNA and nick sgRNA target sites with PAMs facing inwards (PAM-In configuration) and outwards (PAM-Out configuration) at *ATP1A1* exon 4. **e** PCR-based genotyping of *ATP1A1* exon 4 from single cell-derived K562 clones targeted with PAM-out and PAM-in configurations using different nick sgRNAs. Single cell-derived clones were isolated in methylcellulose-based semi-solid RPMI media supplemented with 0.5 μM ouabain and genomic DNA was harvested after co-selection. *n* = 17 single cell-derived clones for each condition from one experiment. Ins, insertion. Del, deletion.

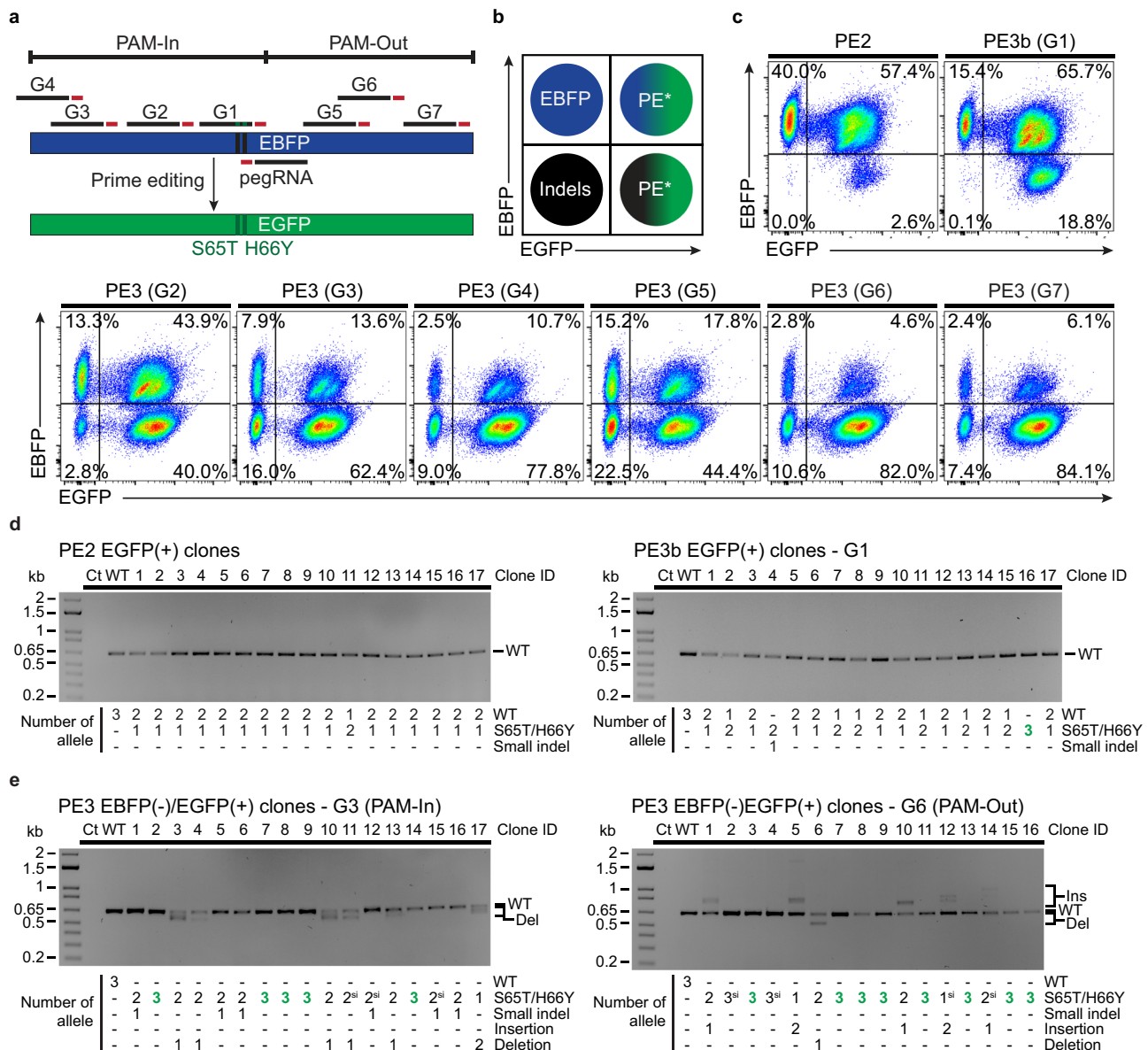

**Fig. 6 | Assessing the impact of complementary-strand nick locations on prime editing outcomes. a** Schematic representation of the EBFP to EGFP reporter system using different nick sgRNAs with PAMs facing outwards (PAM-Out) or inwards (PAM-In) towards the pegRNA. **b** Schematic of the FACS-based quantification of EBFP to EGFP conversion via PE. In K562 cells homozygous for EBFP integration at *ATP1A1* (triallelic), FACS-based analysis reports PE outcomes indirectly; (i) EBFP(+)/EGFP(+) cells result from monoallelic or biallelic PE, (ii) EBFP(−)/EGFP(+) cells are generated by either triallelic PE or combinations of mono- and biallelic PE along with indels*, (iii) EBFP(−)/EGFP(−) occur from indels on the three copies of the reporter. Indels are defined broadly in this context as any edits that inactivates the reporter. **c** FACS-based quantification of EBFP to EGFP conversion via PE after co-selection. K562 cells stably expressing the EBFP reporter from the *ATP1A1* locus were transfected with PE3max vectors targeting *ATP1A1* exon 4 (pegRNA-Q118R_v1) and EBFP (epegRNA). Cells were treated with 100 μM ouabain starting 3 days post-transfection until all non-resistant cells were eliminated. *n* = 3 independent biological replicates performed at different times with equivalent results (see

Supplementary Figs. 22, 23). **d** PCR-based genotyping of EGFP(+) single cell-derived K562 clones targeted with the PE2 or PE3b strategy. Single cell-derived clones were isolated in methylcellulose-based semi-solid RPMI media supplemented with 100 μM ouabain and genomic DNA from EGFP(+) clones was harvested after co-selection. The number of precise prime edited alleles was determined from BEAT Sanger sequencing trace analysis and small indels were analysed with DECODR. *n* = 17 single cell-derived clones for each condition from one experiment. **e** PCR-based genotyping of EBFP from single K562 cell-derived clones after sorting for EBFP(−)/EGFP(+) cells. The number of precise prime edited alleles was determined from BEAT Sanger sequencing trace analysis and small indels were analysed with DECODR. Larger insertions and deletions are indicated, and homozygous single cell-derived clones are highlighted in bold and green. PE alleles harboring pegRNA scaffold incorporation are indicated (si). *n* = 17 and 16 single cell-derived clones from one experiment for the PE3-G3 and PE3-G6 conditions, respectively. Ins, insertion. Del, deletion.

suggesting that EBFP(-)/EGFP(+) cells likely result from PE on all three copies of the reporter. Indeed, co-selected EGFP(+) clones isolated in methylcellulose media were mostly from monoallelic events for PE2, while mono- and biallelic clones occurred via PE3b. Neither large nor short indels were detected in these clones, except for one with a short indel in the PE3b samples (Fig. 6d). In sharp contrast, most cells were

modified at the three copies of the reporter after co-selection when the PE3 strategy was used. This shift was accompanied by the appearance of a population of EBFP(−)/EGFP(−) cells indicating triallelic inactivation of the reporter (Fig. 6c and Supplementary Fig. 23). Hence, the EBFP(−)/EGFP(+) population is likely a mix of cells with triallelic PE and cells with mono- or biallelic PE combined with indels. To test this

hypothesis directly, we FACS-sorted, cloned, and genotyped EBFP (-)/EGFP(+) cells after PE3-based editing and co-selection. Many clones were homozygous for the EBFP to EGFP conversion, but different insertions and deletions byproducts were observed in clones with mono- or biallelic PE (Fig. 6e). Again, large insertions were observed with the PAM-out strategy and a few clones had larger deletions with the PAM-in mode (Fig. 6e). The various complementary-strand nicks tested yielded the same trend with larger insertions in PAM-out and larger deletions with PAM-in configurations (Supplementary Fig. 24). Overall, our results demonstrate that despite the decrease in product purity observed in the cell population, using a secondary nick (PE3) drastically increases multiallelic editing in individual cells. In addition, the type of ssDNA overhangs created by the presence of concomitant nicks affects the type of editing byproducts observed at the target site.

## Discussion

The efficiency of PE fluctuates markedly according to the particular edit, target loci, and cell type[1–3,7]. Despite recent key improvements, it still requires substantial level of optimization to use the technology successfully[7,64]. In this work, we describe a marker-free co-selection strategy that substantially and stably improved the outcome of PE for every pegRNAs tested. The extent of enrichment varies with different pegRNAs, but the strategy greatly facilitates the enrichment of both cell pools and homozygous clones to study the impact of variants on gene function. Due to the high efficiency of deriving novel cell lines, multiple independent cell lines can be analysed to ensure that any unexpected result is not from a clonal artifact[57]. The use of well-characterized dominant gain-of-function mutations[36,37,65,66] in *ATP1A1* introduces minimal modifications to the human genome without detectible effect on cell fitness nor impact on ATP1A1 steady-state expression levels. The two main mutations used during this work, T804N and Q118R, have been extensively studied and these mutant enzymes function normally, as shown by 86Rb+ uptake and ATP hydrolysis assays[35,37]. For functional studies, any potential impact of the modified sodium-potassium pump on the pathway under study can be easily controlled for using a selected control cell line containing only the *ATP1A1* mutations. Typically, the most efficient strategy for the selection of edited cells is targeting endogenous genes as opposed to the use of exogenous markers[19,22,23,67]. Nonetheless, alternative methods have been described to enrich for edited cells, such as puromycin-based selection strategy and the use of surrogate reporters[10,18]. Selecting cells that express all PE components with a selection marker is not sufficient for robust enrichment, while FACS-based cell sorting using exogenous gene markers could be prohibitive for some cell types due to cellular stress[10,18].

An improvement made to our previously described strategy[19] is the option to perform successive rounds of co-selection to install multiple genetic modifications sequentially to derive cell lines. Multiple edits can be engineered at the pool level followed by isolation of clones of interest or new modifications can be added to a previously characterized clone enabling the rapid creation of isogenic cell lines. As the method is portable to CRISPR nucleases, base editors, and prime editors, different types of genomic changes can be implemented. Potential applications include (i) gene complementation with variants coupled to endogenous knock out, (ii) epitope tagging followed by creation of point mutations, and (iii) rescue experiments to confirm causality between edit and phenotype. To create heavily modified human cell lines for applications such as synthetic biology, one could couple our approach with orthogonal co-selection approaches based on auxotrophy or toxins[22,67]. As co-selection is now a proven and established strategy, it appears that other pathways and systems such as MAPK signaling[68], proteasome[69], translation[70], and other tissue-specificities[71] could form the basis of future applications of the methodology.

While the complete mechanism of action of PE remains to be fully elucidated, DNA mismatch repair (MMR) was identified as a genetic determinant of PE outcome. Specifically, MMR inhibits PE and promotes the formation of undesired indel byproducts[7,43]. In the context of co-selection, clonal analysis revealed a high degree of correlation between editing at *ATP1A1* and modification at three different positions within *MTOR*. As such, only one out of 88 selected clones did not display any gene modification (PE or indels) at *MTOR* and only 5 clones did not have at least one pegRNA-specified allele. This clearly indicates that cells proficient at completing one PE3-driven genomic manipulation have an increased probability of completing a second event at a distinct locus. Moreover, approximately a third (31/88) of selected clones were found to be homozygous for the specified mutations at the *MTOR* locus in the triploid K562 cell line. Interestingly, clones triallelic for the I2017T mutation were observed at a frequency of 26% (7/27) even though PE levels in the pool of selected cells only reached 39%. Thus, when active in a cell, PE3 is prone to edit more than one allele of its target gene. Taken together, our data suggest that only a fraction of cells in an asynchronous population are competent for PE. Although this observation warrants further investigation, a detailed mechanistic understanding of the PE reaction may help to explain these results. A confounding factor in this analysis may stem from *MTOR*'s essentiality, but we did not observe any growth delay in either pools of cells or when clones developed into colonies in methylcellulose for the various genotypes. These observations were also made in the EBFP to EGFP reporter system.

We selected the PE3 strategy to initiate co-selection because it is more active than PE2 in human cells[1,7]. This reduced the time required to obtain a fully resistant population of cells. However, PE3 comes at the expense of product purity and is linked to the formation of indels[1,7,13]. Genotyping of single cell-derived clones allowed us to better describe the different types of PE3 byproducts. PE3 designs with PAMs facing outwards with respect to each other (PAM-out) are predicted to create DSBs with 3' ssDNA protruding ends that are similar to a paired nickase strategy[72]. As such, the major repair byproducts are direct repeats of the sequences found within the overhangs[59,73]. Multiple repeated sequences could be formed by consecutive rounds of repair and nicking since the target sites are still present in the tandem duplications[73]. Mechanistically, fill-in synthesis of the 3' protruding ends by Polα-primase can account for these tandem duplications in a pathway that requires 53BP1[60–62]. Notably, a CRISPRi screen revealed that *TP53BP1* knockdown drastically reduces the frequency of tandem duplications during PE3 (PAM-out)[7] further suggesting that editing byproducts are linked to the DSB repair response resulting from presence of concomitant nicks. Taken together, these data suggest that a competition between PE3 and NHEJ determines the overall efficiency of prime editing. Hence, delaying rapid and predominant NHEJ-mediated repair may improve PE3 yields and purity. DNA end protection mediated by 53BP1[61,74] could modulate PE3 outcome as 53BP1-deficient cells have delayed repaired kinetics that can lead to a switch in the choice of repair pathway[75]. However, we failed to detect a change in repair outcome when using i53[63], the genetically encoded 53BP1 inhibitor in this context. In contrast, when using PE3 in PAM-in configuration, deletions were more prevalent and larger as observed with Cas9 dual nickases creating 5' ssDNA protruding ends[59]. One could predict the formation of translocations between *ATP1A1* and GOI if DSB repair pathways are involved in PE3. Indeed, we could detect rare occurrences of translocations during co-editing via PE3 using a previously described nested PCR-based approach[76,77]. While this assay is not quantitative, detection was much more robust when swapping the prime editor expression vector for wild-type SpCas9. This corroborates the findings made when comparing nucleases vs. paired-nickases in this system[76].

Finally, we note that combining optimized epegRNA with maximal prime editor expression, we were able to isolate homozygous K562 clones for the *MTOR*-F2108L mutation using the PE2 strategy. This demonstrates that, while infrequent, homozygous clones can also be

isolated without using a secondary nick. No unwanted byproducts were observed in this context, providing a compromise between product purity and efficiency (Supplementary Table 2).

Taken as a whole, the various strategies presented here should further streamline the incorporation of marker-free genetic changes in human cells while being compatible with future improvements in PE technology.

## Methods

### Cell culture and transfection
K562 cells were obtained from the ATCC (CCL-243) and maintained at 37 °C under 5% $CO_2$ in RPMI medium supplemented with 10% FBS, penicillin−streptomycin, and GlutaMAX. HeLa S3 and U2OS cells were obtained from the ATCC (CCL-2.2 and HTB-96) and maintained at 37 °C under 5% $CO_2$ in DMEM medium supplemented with 10% FBS, penicillin−streptomycin, and GlutaMAX. Cells were routinely tested for the absence of mycoplasma contamination. Ouabain octahydrate (Sigma) was dissolved at 5 mg/ml in hot water, working dilutions were prepared in water and stored at −20 °C. Ouabain should be handled carefully per manufacturer's recommendations following standard safe laboratory practices. Rapamycin (Cayman Chemicals) was dissolved at 10 mg/ml in DMSO, working dilutions were prepared in water and stored at −20 °C. AZD8055 (STEMCELL Technologies) was dissolved at 10 mM in DMSO, working dilutions were prepared in water and stored at −20 °C. DMSO alone was diluted in water and used as vehicle control. K562 cells ($2 \times 10^5$ cells/transfection) were transfected with an Amaxa 4D-nucleofector™ (Lonza) using the SF nucleofection kit (program FF-120) per manufacturer's recommendations. HeLa S3 cells ($2 \times 10^5$ cells/transfection) were transfected using the SE nucleofection kit (Standard HeLa program CN-114 and High efficiency HeLa S3 program DS-150). U2OS cells ($3-4 \times 10^5$ cells/transfection) were transfected using the SE nucleofection kit (program DN-100). Amounts of DNA used for each experiment can be found in the Supplementary material section.

Cells were treated with the indicated concentrations of ouabain starting 3 days post-nucleofection until all non-resistant cells were eliminated. Typical co-selection for prime editing requires 10−14 days for K562 cells, and 14−17 days for HeLa S3 and U2OS cells. Timing can vary according to initial modification rates at *ATP1A1*, and we recommend using the high efficiency HeLa S3 program (DS-150) for HeLa S3 cells. Simultaneous co-selection and cloning was performed in methylcellulose-based semi-solid RPMI medium supplemented with 100 μM ouabain for 10 days. K562 cell lines constitutively expressing EBFP or EGFP from the *AAVS1* safe-harbor were generated as previously described[78]. Briefly, K562 cells were transfected with 400 ng eSpCas9(1.1)_No_FLAG_AAVS1_T2 (Addgene 79888) and 800 ng of either AAVS1_Puro_hPGK1_EBFP_Donor (Addgene 178089) or AAVS1_Puro_hPGK1_EGFP_Donor (Addgene 178088). Three days post transfection, cells were cloned in methylcellulose-based semi-solid RPMI medium supplemented with 0.5 μg/ml puromycin for 10 days. Clones were picked and expanded in 96 wells with 0.5 μg/ml puromycin before genomic DNA extractions. Clones of interest were then expanded in 24-well and 6-well plates prior to FACS analysis. To establish the K562 cell line constitutively expressing EBFP from *ATP1A1* intron 17, cells were transfected with 350 ng pX330_ATP1A1_G7 (Addgene 173204) and 700 ng ATP1A1-T804N_hPGK1_EBFP_Donor (Addgene 187454). Three days post transfection, cells were cloned in methylcellulose-based semi-solid RPMI medium supplemented with 0.5 μM ouabain for 10 days. Clones were picked and expanded in 96 wells with 0.5 μM ouabain before genomic DNA extractions. Clones of interest were then expanded in 24-well and 6-well plates prior to FACS analysis.

### Genome editing vectors and reagents
Adenine base editing (ABE) and prime editing (PE) experiments were performed with pCMV_ABEmax[79] (Addgene 112095), ABE8e[80] (Addgene 138489), ABE8e (TadA-8e V106W)[80] (Addgene 138495), pCMV-PE2[1]

(Addgene 132775), pCMV-PEmax (Addgene 174820), pCMV-PEmax-P2A-hMLH1dn (Addgene 174828), pU6-pegRNA-GG-acceptor[1] (Addgene 132777), and pU6-tevopreq1-GG-acceptor (Addgene 174038). These vectors were gifts from David R. Liu. U6-pegRNA cassette expressing the *ATP1A1*-Q118R and T804N cassettes were cloned upstream of the U6-RFP-acceptor cassette to create *ATP1A1*_G4_Q118R_Dual_pegRNA (Addgene 173199) and *ATP1A1*_G6_T804N_Dual_pegRNA (Addgene 173200). For co-selection, pegRNAs targeting GOI were cloned in the dual pegRNA vectors using the same protocol[1]. For PE3 nick sgRNAs, sgRNAs were cloned into an in-house pUC19-U6-BbsI-sgRNA vector. The U6-*ATP1A1* nick sgRNA cassettes were cloned upstream of the U6-BbsI-sgRNA cassette to create *ATP1A1*_G3_Dual_sgRNA (Addgene 173202) and *ATP1A1*_G8_Dual_sgRNA (Addgene 178104). For PE3 co-selection, nick sgRNAs targeting the genes of interest were cloned into the BbsI sites of the dual nick gRNA vectors. The U6-*ATP1A1* sgRNA (G2) cassette was also cloned upstream of the U6-BbsI-sgRNA cassette to create *ATP1A1*_G2_Dual_sgRNA (Addgene 173201) for ABE co-selections. When required, DNA sequences for the guides were modified at position 1 to encode a G, owing to the transcription requirement of the human U6 promoter. To assess the impact of 53BP1 inhibition on PE3-mediated large insertions, experiments were performed with pcDNA3-Flag::UbvG08 I44A, deltaGG (Addgene 74939), a gift from Daniel Durocher, and pEF1a-hMLH1dn (Addgene 174824). All sgRNA, pegRNA, and ssODN sequences used during this study are provided in the Supplementary material section.

For nuclease-based co-selections, sgRNA sequences were cloned into eSpCas9(1.1)_No_FLAG (Addgene 79877)[78], eSpCas9(1.1)_No_FLAG_*ATP1A1*_G3_Dual_sgRNA1[19] (Addgene 86613) or pX330-U6-Chimeric_BB-CBh-hSpCas9[81] (Addgene 42230), a gift from Feng Zhang. Plasmid donor sequences used for *ATP1A1* gene knock-in were cloned by restriction cloning in a pUC19 backbone. All plasmid donor sequences contained short homology arms (<1 kb) and they were modified to prevent Cas9 cleavage. To repurpose the mTORC1 signaling reporter mVenus-TOSI previously developed for mouse[55], the N-terminal residues (1-82) of the human *PDCD4* gene were codon optimized using the GenSmart™ codon optimization tool, synthesized as a gBlocks™, and cloned into ATP1A1_T804N_hPGK1_mScarlet-I_Donor (Addgene 173207) upstream of the mScarlet-I-NLS cassette using AflII and NcoI. For puromycin selection, the TOSI_mScarlet-I-NLS cassette was transferred to *AAVS1*_Puro_PGK1_3×FLAG_Twin_Strep[78] (Addgene plasmid 68375). Desalted ssODNs (see Supplementary material) were synthesized as ultramers (IDT) at a 4 nmol scale. A series of vectors described in this work (see Supplementary material) have been deposited to Addgene.

### Flow cytometry
The percentage of fluorescent cells was quantified using a BD LSRII flow cytometer using BD FACSdiva v6.1.2 software, and $1 \times 10^5$ cells were analyzed for each condition. When HDR donor plasmids harboring a human *PGK1* promoter were used to target fluorescent gene cassettes to the *ATP1A1* locus, cells were cultured for 21 days to eliminate residual expression from non-integrated reporter cassettes. For mTORC1 signaling assays, cells were treated with the indicated concentration of rapamycin or AZD8055 for 24 h before analysis. Bulk population of EBFP(−)/EGFP(+) cells were sorted using a BD FACS Aria Fusion cytometer. Serial dilution cloning was performed in 96-well plates to isolate single-cell derived clones. Flow cytometric data visualization and analysis was performed using FlowJo (v10 and v11) software (Tree Star). The gating strategies used during this study are provided in Supplementary Fig. 25.

### Genotyping
Genomic DNA was extracted with QuickExtract DNA extraction solution (EpiCentre) following manufacturer's recommendations. Primers used in this study and the PCR product sizes are provided in the

Supplementary material section. All PCR amplifications were performed with 30 cycles of amplification. Sanger sequencing was performed on PCR amplicons to quantify the percentage of edited alleles using BEAT[44], TIDE[45], and TIDER[56]. Phusion and Q5 high-fidelity DNA polymerases were used for PCR amplification. Kapa-HiFi polymerase was used for out-out PCRs. To detect targeted integration, out-out PCRs were performed with primers that bind outside of the homology regions of the donor plasmids. Wild-type K562 genomic DNA was used as a negative control for all out-out PCRs. The uncropped scans of all gels from this study are provided in Supplementary Fig. 26.

## Cell proliferation assays
Following ouabain selection, bulk populations of K562 cells harboring *ATP1A1* mutations were seeded in technical triplicate at 75,000 cells/ml in a 6-well plate with 4 ml of RPMI medium supplemented with 10% FBS, penicillin–streptomycin, and GlutaMAX. Growth was monitored for 7 days using a Neubauer counting chamber and non-viable cells were excluded based on Trypan blue staining.

## Western blot analysis
Briefly, $1.2 \times 10^6$ K562 cells were pelleted and resuspended in 100 μl hot lysis buffer (1% SDS, 10 mM Tris-HCl, pH 8.0) supplemented with 1X Halt™ protease and phosphatase inhibitor cocktail (Thermo Scientific, catalog # 78446). Samples were sonicated 20 times with 3 s ON and 3 s OFF cycles. Protein extracts were diluted in Laemmli buffer and heated at 70 °C for 10 min before loading on a 7.5% polyacrylamide Mini-PROTEAN TGX Stain-Free™ gel (BioRad, catalog # 4568023). Nitro-cellulose membranes were blocked with 5% milk dissolved in PBST and incubated overnight at 4 °C using an anti-α-Na$^+$/K$^+$ ATPase antibody (Invitrogen, catalog # MA3-928) diluted 1:1000 in PBST 5% milk, and 1 h at room temperature using an anti-mouse secondary antibody (Cell Signaling Technology, catalog # 7076 S) diluted 1:5000 in PBST 5% milk. Membranes were subsequently incubated for 1 h at room temperature with an anti-α-Tubulin antibody (Santa Cruz Biotechnology, catalog # sc-32293) diluted 1:3000, followed by 1 h at room temperature using the same anti-mouse secondary antibody. The uncropped scans of all blots from this study are provided in Supplementary Fig. 26.

## PCR-based translocation detection
Translocations were detected as previously described using a nested PCR strategy on genomic DNA at breakpoint junctions[76,77]. As positive controls, K562 cells were transfected with a nuclease-expression vector (Addgene 42230) and dual pegRNAs- and sgRNAs-expression vectors targeting *ATP1A1* and a gene of interest (*EMX1* or *RUNX1*). For PE3 conditions, genomic DNA samples from PE3 co-selections (Fig. 1) were used to detect the presence of translocations. The first step of PCR was performed with 25 cycles of amplification using primers on each side of the predicted translocation junctions. The PCR products were diluted 1:10 in nuclease-free water and used for a successive step of nested PCR with 25 cycles of amplification using primers binding within the first product. Sanger sequencing was performed after nested PCRs with primers on each side of the translocation junctions. The uncropped scans of all gels from this study are provided in Supplementary Fig. 26.

## Reporting summary
Further information on research design is available in the Nature Research Reporting Summary linked to this article.

## Data availability
Source data for the figures are provided as a Source Data file. For raw sequencing reads, the International Nucleotide Sequence Database Collaboration (INSDC) repositories only accepts Next Generation sequence reads (Sequence Read Archive) and assembled genome sequences. Hence, the recommended repositories do not support raw

Sanger sequencing reads. All raw Sanger sequencing reads generated in this study are available on request from the corresponding author [YD] without any restrictions of access. All vectors generated in this study have been deposited to Addgene (https://www.addgene.org/browse/article/28219935/). Source data are provided with this paper.

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

## Acknowledgements

This study was supported by a grant from the Natural Sciences and Engineering Research Council of Canada (RGPIN-2014-059680). S.L. holds a Frederick Banting and Charles Best Canada graduate scholarship from CIHR. A.D. holds a doctoral training award from the Fonds de recherche du Québec en Santé (FRQS). D.A. holds a Vanier Canada graduate scholarship from CIHR. Salary support was provided by the Fonds de la Recherche du Québec-Santé (FRQS) to Y.D. We thank the staff at the flow cytometry core facility of the CHU de Quebec Research Center for their training and assistance.

## Author contributions

Conceptualization, S.L. and Y.D.; methodology, S.L., D.M., J.-P.F., C.G., A.D., A.L., E.B., D.A., and Y.D.; investigation, S.L., D.M., J-P.F., C.G., A.D., A.L., E.B., D.A., and Y.D.; writing original draft, S.L.; writing, review and editing, S.L., and Y.D.; supervision, Y.D.; funding acquisition, Y.D. All authors reviewed the manuscript and approved its final version.

## Competing interests

The authors declare no competing interests.
