## [Peer Review File · Nature Communications]

Reviewers' Comments:

Reviewer #1:

Remarks to the Author:

Review of Levesque S. et al.

The Achilles' heel of the current genome editing technologies for cell engineering is their poor overall efficiency to install desired mutations and/or indels. Several studies in the field have tackled this challenge by developing selection or co-selection systems using ectopic or endogenous genes that would allow for the enrichment of the edited cells (1, 2, 3, 4, 5, 6, 7).

The authors of the reviewed manuscript had previously published a method to co-select for simultaneous genome edits (4) and the same co-selection system has been used to enrich for base editing events (8). Briefly, precise mutations in the ATP1A1 gene render cells resistant to ouabain, a plant-derived toxin that inhibits the protein product of the gene, a sodium-potassium ion pump. In the current manuscript, the authors (1) identified mutations in the ATP1A1 gene that confers intermediate resistance to ouabain that allows for a two-step selection; and (2) used the selection system for transgene expression at the ATP1A1 locus. Moreover, the authors extended their co-selection system to (3) precise genome editing by HDR-mediated integration of a DNA cassette in ATP1A1 locus as well as (4) Prime Editing (PE).

Overall, I acknowledge and recognize the work done of the authors. However, I find the presented study provides little conceptual advances and technological benefits over the currently existing methods previously published. Furthermore, the manuscript lacks the essential information about the reproducibility of experiments and data presentation offers a major opportunity for improvements to reader's benefit. Please see below my points of criticism. Altogether, I do not recommend this work in its current premature form for scientific publication.

Major

- The manuscript presents no major advances over previous study from the same lab (4) and other technologies for cell line engineering, questioning the impact of this work. Prime-editing co-selection is a novel observation but expected considering previous data on KO, HDR, BE co-selection using the same system (4, 8). Moreover, it is not clear from the discussion the practical benefit PE co-selection has over HDR and BE based co-selection.

- Throughout the manuscript, I frequently find no information about the experimental replicates, statistics, etc. Does it mean the experiments were done at the N=1? If yes this questions the reproducibility of the presented data

- Safety profiles and molecular/cellular consequences of (a) the mutations introduced in the ATP1A1 or even (b) transgene expression from the ATP1A1 locus have not been addressed or even attempted. I find difficult to understand why such a fundamental and critical information was not included in the manuscript.

- The transgene integration strategy offers no clear advancement respect to previously published technologies (6) and it actually seems inferior to them. In particular (a) The authors don't explain why a wild type allele is still present in the cell pool after Oubain mediated selection that should kill cells with wild type alleles or alleles with small indels. No data is provided about multiallelic vs monoallelic integrations (b) The presence of the transgene in opposite orientation to ATP1A1 may interfere with ATP1A1 expression but the authors don't provide any info on ATP1A1 expression after transgene integration

- Longitudinal expression studies not shown using the expression method from the ATP1A1 locus with different cell types (including primary and iPSCs throughout differentiation). I find no rationale for using the term "safe harbor locus" also considering previous published description of safe harbor site (9)

- Given that authors present their findings as a method to engineer cell lines, I find critical to consider the safety of using the toxic substance which has not been presented and discussed.

Minor

- Complicated and nonintuitive presentation of the data (Figure 1 and 2). As suggested before the presentation of bar charts (instead of FACS plot based on a single experiment) with proper statistical analysis based on separate experiments would strengthen the quality of the manuscript.
- Details of the HDR experiment in Figure 1c not provided and it is difficult to explain the reason of a very low efficiency of ssODN mediated HDR without selection
- Lack of CRISPR editing analysis using Amplicon-Seq. This analysis would greatly benefit the quality of the article.
- Imbalanced introduction with a main description and focus on Prime Editing regardless limited focus on prime editing in the manuscript. The manuscript would benefit from an expansion of the Prime Editing part. One suggestion would be to provide experimental evidence with i53 cp-expression of the interesting proposed 53BP1 involvement in PE pathway as suggested in line of text 420.
- Enrichment after Oubain selection of PE events at ATP1A1 leads to around 85% of precise, wanted edits. What is the type of edits present in the remaining resistant cells after PE?

1 Kim, Heesun, et al. "A co-CRISPR strategy for efficient genome editing in *Caenorhabditis elegans*." *Genetics* 197.4 (2014): 1069-1080.

2 Flemr, Matyas, and Marc Bühler. "Single-step generation of conditional knockout mouse embryonic stem cells." *Cell reports* 12.4 (2015): 709-716.

3 Shy, Brian R., et al. "Co-incident insertion enables high efficiency genome engineering in mouse embryonic stem cells." *Nucleic acids research* 44.16 (2016): 7997-8010.

4 Agudelo, Daniel, et al. "Marker-free coselection for CRISPR-driven genome editing in human cells." *Nature methods* 14.6 (2017): 615-620.

5 Ewen-Campen, Ben, and Norbert Perrimon. "ovoD co-selection: A method for enriching CRISPR/Cas9-edited alleles in *Drosophila*." *G3: Genes, Genomes, Genetics* 8.8 (2018): 2749-2756.

6 Li, Songyuan, et al. "Universal toxin-based selection for precise genome engineering in human cells." *Nature communications* 12.1 (2021): 1-14.

7 Reuven, Nina, et al. "CRISPR Co-Editing Strategy for Scarless Homology-Directed Genome Editing." *International Journal of Molecular Sciences* 22.7 (2021): 3741.

8 Billon, Pierre, et al. "CRISPR-mediated base editing enables efficient disruption of eukaryotic genes through induction of STOP codons." *Molecular cell* 67.6 (2017): 1068-1079.

9 Sadelain, Michel, Eirini P. Papapetrou, and Frederic D. Bushman. "Safe harbours for the integration of new DNA in the human genome." *Nature reviews Cancer* 12.1 (2012): 51-58.

Reviewer #2:

Remarks to the Author:

In this manuscript the authors reported a marker-free coselection strategy of prime editing based on dominant alleles of ATP1A1 Na⁺/K⁺ ATPase. This is an extension of the authors' previous study of coselection of CRISPR editing (*Nat. Methods* 2017). This method improves prime editing efficiency 1.5–10-fold in K562 and HeLa cells. The authors engineered homozygous cell lines with cancer-associated or drug resistant MTOR mutations and used a MTOR reporter knock-in to display altered mTORC1 signaling. This approach has potential to increase the PE efficiency in cell lines. Overall this is a significant manuscript which will advance the improvement of PE in cells. Several questions need to be addressed in the revision.

Major:

1. A major concern of this study is that the authors did not examine the off-target (OT) effects of Cas9 (ATP1A1 HDR) and PE, particularly when co-selection may enrich the co-editing at OT sites. The authors should quantify the OT effects of Cas9 and PE after co-selection.

2. The authors noted that K562 is triploid. The copy number of ATP1A1 and mTOR needs to be clarified or measured.

Minor:

- PE can induce low levels of deletion between pegRNA and nicking sgRNA target sites. This is not assayed in the manuscript. Were the indels in Figs. 3 and 4 all small indels?

Point-by-point response to referees

*Please note that we have highlighted the changes made to the manuscript since the original submission using blue font for the text.

Reviewer #1:

We thank the reviewer for acknowledging our work, more specifically, for recognizing the novelty and usefulness of our work on prime editing (PE). Overall, we understood that the manuscript would have benefited from an expansion of the PE part over nuclease-based applications. We agree with the reviewer and have now refocused the manuscript on the PE aspect. Another major criticism was the complicated, nonintuitive, and incomplete presentation of the data. We have made several adjustments to the figures to improve clarity and addressed the issue about reproducibility.

Notable improvements to our manuscript concern the mechanism of PE in human cells. Clonal analysis (we have isolated 485 single cell-derived clones) revealed the impact of the complementary-strand nick (PE3, PE5) (i) on the rate of homozygosity, and (ii) on product purity. In particular, we have observed a direct relationship between the position of the 2nd nick relative to the prime editing site (PAM-in vs PAM-out configuration) that drives the formation of either tandem duplications or deletions at the target site. This is a previously unrecognized on-target byproduct of PE that should be accounted for when designing PE experiments. This is now described in Figures 3-5 of the manuscript.

Major comments:

- The manuscript presents no major advances over previous study from the same lab (4) and other technologies for cell line engineering, questioning the impact of this work. Prime-editing co-selection is a novel observation but expected considering previous data on KO, HDR, BE co-selection using the same system (4, 8). Moreover, it is not clear from the discussion the practical benefit PE co-selection has over HDR and BE based co-selection.

While co-selection for PE may have been expected based on previous data with other CRISPR-based editing systems, such a system had to be implemented and tested to demonstrate if it could be useful or not for cell line engineering. We believe that the data presented in this revised manuscript clearly does so. In addition, to the best of our knowledge, this is the first demonstration that a co-selection strategy can be used for multiple steps of editing.

Each CRISPR–Cas nucleases, base editors, and prime editors systems have pros and cons and they must be selected according to the application irrespective of co-selection (Anzalone et al, Nature Biotechnology, 2020 PMID: 32572269). For example, “in most cases indels still represent the majority of edited products with Cas nuclease-initiated HDR” *ibid*. This is an observation we have previously reported during HDR-driven co-selection (Agudelo et al, Nature Methods, 2027 PMID: 28417998). It is currently assumed that the ratio of precise edits/indels is higher for PE than HDR-based techniques and we now present a side-by-side comparison in Supplementary Fig. 8 of the revised manuscript. The reviewer will also note that throughout our work, we measured the levels

of indels versus PE. In the vast majority of cases PE is dominant over the creation of indels during co-selection. This represents, in our opinion, a practical benefit of PE co-selection over HDR co-selection when creating “small” edits.

- Throughout the manuscript, I frequently find no information about the experimental replicates, statistics, etc. Does it mean the experiments were done at the N=1? If yes this questions the reproducibility of the presented data

To address concerns related to experimental replicates, we merged all data from the supplemental figures to the main figures and we presented the data throughout the manuscript as dot plots from independent replicates performed at different times. We agree that this is much better than our previous approach where we presented the different repeats of an experiment in the supplemental material section.

- Safety profiles and molecular/cellular consequences of (a) the mutations introduced in the *ATP1A1* or even (b) transgene expression from the *ATP1A1* locus have not been addressed or even attempted. I find difficult to understand why such a fundamental and critical information was not included in the manuscript.

(a) The coselection strategies presented as part of this work were inspired by a long history of mutagenesis and biochemical analysis on the Na^+/K^+ ATPase. While we cannot exclude that the mutations created are not totally neutral, we note that ouabain resistance mutations such as the ones we describe, are naturally occurring in metazoans (Ujvari *et al.* PNAS 2015). Moreover, the two main mutations used during this work, T804N and Q118R, have been extensively studied and these mutant enzymes function normally, as shown by $^{86}\text{Rb}^+$ uptake and ATP hydrolysis assays (Price *et al.* JBC 1990, and Burns and Price, JBC 1993). Considering that the *ATP1A1* gene is essential, any functional consequences would prevent robust cellular growth in the presence or absence of ouabain, which was not observed during this work. This information was presented in our previous work and has been added to the current manuscript in the discussion section.

We also want to highlight that any impact of these mutations can easily be controlled for with a PE-*ATP1A1* only control (Fig. 2 and Discussion). Furthermore, several laboratories have used our method successfully in near-haploid, diploid and polyploid cells without obvious limitations.

(b) The reviewer is correct to point out that we neither used the correct standard nor the correct terminology when using the term “safe harbor locus”. We have eliminated this section of the manuscript. This strategy was nevertheless useful for our work when we established (i) the mTOR reporter cell line, (ii) the EBFP to EGFP reporter system. In the current version of the manuscript, we only mentioned that this “intron nesting” strategy was successful for these applications and made no forward-looking statements. That said, we have found that expression remains stable and does not fluctuate over time in pools of selected cells.

- The transgene integration strategy offers no clear advancement respect to previously published technologies (6) and it actually seems inferior to them. In particular (a) The authors don't explain why a wild type allele is still present in the cell pool after Oubain mediated selection that should kill cells with wild type alleles or alleles with small indels. No data is provided about multiallelic vs monoallelic integrations (b) The presence of the transgene in opposite orientation to *ATP1A1* may interfere with *ATP1A1* expression but the authors don't provide any info on *ATP1A1* expression after transgene integration.

As noted above, we agree with the reviewer and have removed these data/discussion from the manuscript. We can still offer some answers to the reviewer's questions regarding this TI strategy:

(a) Considering that we are installing dominant gain of function modifications at *ATP1A1*, a single edited allele is sufficient to confer ouabain resistance. We have performed out-out PCR based assays in single cell-derived clones and have observed that most clones are homozygotes for TI for both the mTOR and the EBFP to EGFP reporters (Supplementary Figs. 14 and 21).

(b) It is a possibility that *ATP1A1* expression is decreased upon TI of an expression cassette within one of its introns. We have not addressed this question directly but in view of the essentiality of this gene, we would have expected to see an impact on growth, which we didn't (even in homozygotes with 3 alleles targeted). We note that we used a relatively weak promoter (PGK1) and a strong polyA signal in order to mitigate this potential problem.

- Longitudinal expression studies not shown using the expression method from the *ATP1A1* locus with different cell types (including primary and iPSCs throughout differentiation). I find no rationale for using the term "safe harbor locus" also considering previous published description of safe harbor site (9).

As noted above, we agree with the reviewer and have removed these data/discussion from the manuscript.

- Given that authors present their findings as a method to engineer cell lines, I find critical to consider the safety of using the toxic substance which has not been presented and discussed.

A safety warning from the manufacturer has been added to the material section. We have also described how to prepare stock solutions and use ouabain in the methods section.

Minor comments:

- Complicated and nonintuitive presentation of the data (Figure 1 and 2). As suggested before the presentation of bar charts (instead of FACS plot based on a single experiment) with proper statistical analysis based on separate experiments would strengthen the quality of the manuscript.

To address the reviewer's concerns regarding data presentation, all data are now presented throughout the manuscript as dot plots from independent replicates performed at different times instead of separate figures in supplemental material.

- Details of the HDR experiment in Figure 1c not provided and it is difficult to explain the reason of a very low efficiency of ssODN mediated HDR without selection.

These data have now been moved to (Supplementary Fig. 8). We have previously observed higher HDR efficiency when using a WT SpCas9 nuclease, but all coselection for HDR experiments were performed with the high-fidelity eSpCas9(1.1) nuclease, as previously described (Agudelo et al, Nature methods, 2027 PMID: 28417998). The trade-off between on-target and off-target activity with the high-fidelity variant eSpCas9(1.1) is likely responsible for low on-target activity without coselection when using this particular EBFP targeting sgRNA. The experimental conditions are detailed in the Supplementary Information file.

Lack of CRISPR editing analysis using Amplicon-Seq. This analysis would greatly benefit the quality of the article.

- While we agree that Amplicon-Seq could have benefited this work when editing levels were low (0.1% - 5%), the high levels of editing observed during this work are readily detectable with BEAT, TIDE, DECODR, and other Sanger sequencing analysis programs. The percentage of PE alleles were $\geq 5\%$ after coselection with all pegRNAs tested, even when editing levels were undetectable before selection. We have also provided the sequence of several insertions as determined via Sanger sequencing to provide a qualitative view of the data (Figure 3 and Supplementary Figs. 12 and 19).

We would like to emphasize that previous work focusing on large insertions mediated by Cas9 nickases, such as the ones identified in this study, found that Illumina MiSeq underrepresented the insertions counts compared to Sanger sequencing (Bothmer *et al.* Nature Communications, 2017, PMID: 28067217). In our work, Amplicon-Seq may have underestimated large insertion counts and overestimated precise prime editing. This is evident when looking at the gel-based assays we present (Figures 4,5 and Supplementary Fig.13).

- Imbalanced introduction with a main description and focus on Prime Editing regardless limited focus on prime editing in the manuscript. The manuscript would benefit from an expansion of the Prime Editing part. One suggestion would be to provide experimental evidence with i53 cp-expression of the interesting proposed 53BP1 involvement in PE pathway as suggested in line of text 420.

- We agree and we have provided major revisions to the manuscript to focus on prime editing. We have tested the impact of 53BP1 inhibitor i53 (Canny *et al.* Nature Biotechnology, 2018, PMID: 29176614) in K562 and HeLa cells, but we did not observe marked decreases in the generation of unwanted PE-mediated outcomes (Supplementary Fig. 20). The same was true for the use of dominant negative MLH1 used to inhibit MMR (Chen et al. Cell, 2021, PMID: 34653350) (Supplementary Fig. 20).

- Enrichment after Oubain selection of PE events at *ATP1A1* leads to around 85% of precise, wanted edits. What is the type of edits present in the remaining resistant cells after PE?

We provided an extensive analysis of editing outcomes at target sites in the current version of the manuscript for several pegRNAs. When genotyping from pools of selected cells, non-PE alleles are composed of relatively short indels or WT sequences as previously described (Anzalone et al. Nature, 2019, PMID: 31634902). This is also true for *ATP1A1*. Of note, a single edited allele is sufficient to confer ouabain resistance since we are creating dominant gain of function mutations.

By analyzing clones, we were able to ID a novel type of byproducts consisting of larger insertions and deletions when using a secondary nick to increase PE rates (PE3, PE5). These events were likely missed in previous studies relying on analysis of pools of modified cells via amplicon sequencing. We further demonstrated that the relative position of the PAMs dictates whether large insertions or deletions are created. This was observed with several pegRNAs, including *ATP1A1*-targeting pegRNAs.

Reviewer #2:

- We thank the reviewer for their description of our work as “a significant manuscript which will advance the improvement of PE in cells”. Hence, we have refocused the manuscript on the PE technology. Overall, we understood that the reviewer’s main concern was related to the specificity of the approach. While we have not performed a *bona fide* off-target analysis, we have characterized and identified novel undesired byproducts occurring at the target sites during PE. Considering that most studies to date suggest that PE provides a highly specific method of precise genome editing, we hope that the reviewer will agree with the orientation we took during the course of our studies.

Major comments:

1. A major concern of this study is that the authors did not examine the off-target (OT) effects of Cas9 (*ATP1A1* HDR) and PE, particularly when co-selection may enrich the co-editing at OT sites. The authors should quantify the OT effects of Cas9 and PE after co-selection.

Both reviewers suggested to expand PE part of our manuscript over nuclease-based applications, so we focused our work on PE during the revisions. As mentioned above, in most settings, PE is a highly specific process. Considering that most nicks are repaired without indels, the use of prime editor almost completely abolishes off-target editing (Anzalone et al. Nature, 2019, PMID: 31634902). Moreover, high genome-wide specificity of prime editors has recently been reported in plants with barely detectable levels of off-target edits (Jin *et al.* Nature Biotechnology 2022, PMID: 33859403). Schene et al. found that: “Whole-genome sequencing of four prime-edited clonal organoid lines reveals absence of genome-wide off-target effects underscoring therapeutic potential of this versatile and precise gene editing strategy” (Nature Communications, 2020, PMID: 33097693). In addition, Kim et al. conclude that: “In summary, our data, obtained using

the nDigenome-seq method, revealed the high specificity with which precise, PE-mediated genome editing can be achieved.” (Nucleic Acids Research, 2020, PMID: 32941652)

While we agree with the reviewer that specificity is a critical concern for genome editing applications, detecting overt off-target editing with PE after coselection would be very challenging as we do not have known off-target sites with readily detectable levels of off-target edits. We understand that no CRISPR-based technologies will be 100% specific, but we felt that, for applications such as cell line engineering, our efforts would be better invested elsewhere at the moment. Therefore, we hope that the reviewer will agree with us that such studies, while important, are beyond the scope of the current manuscript.

With regards to our co-selection scheme, we want to emphasize that any negative impact related to targeting *ATP1A1* via PE can easily be controlled for with a PE-*ATP1A1* only control (Fig. 2 and Discussion). We also note that we have not observed any impact on cellular growth in the presence or absence of ouabain during this work. We have previously investigated the impact of coselection on nuclease-driven off-targets and we did not detect overt off-target activity at previously characterized off-target sites (Agudelo et al, Nature Methods, 2027 PMID: 28417998). Finally, we cite to several papers showing that our co-selection strategy has proven to be portable to non-homologous end joining (NHEJ), homology directed repair (HDR), and base editing events in near-haploid, diploid and polyploid cells indicating that other laboratories used our method successfully without noticeable problems.

2. The authors noted that K562 is triploid. The copy number of *ATP1A1* and *mTOR* needs to be clarified or measured.

We agree with the reviewer that the copy numbers should be clarified. *ATP1A1* and *MTOR* are both triallelic in K562 cells (Naumann *et al.* Leuk. Res. 2001). This was further confirmed with the genotypes observed in single cell-derived clones. We have clarified this fact in the results section.

Minor comments:

- PE can induce low levels of deletion between pegRNA and nicking sgRNA target sites. This is not assayed in the manuscript. Were the indels in Figs. 3 and 4 all small indels?

We provided an extensive analysis of editing outcomes at target sites in the current version of the manuscript for several pegRNAs. When genotyping from pools of selected cells, non-PE alleles are composed of relatively short indels or WT sequences as previously described (Anzalone et al. Nature, 2019, PMID: 31634902).

By analyzing clones, we were able to ID a novel type of byproducts consisting of larger insertions and deletions when using a secondary nick to increase PE rates (PE3, PE5). These events were likely missed in previous studies relying on analysis of pools of modified cells via amplicon sequencing. We further demonstrated that the relative position of the PAMs dictates whether large insertions or deletions are created. These observations were made with several pegRNAs, including *ATP1A1*-targeting pegRNAs.

Reviewers' Comments:

Reviewer #1:

Remarks to the Author:

The revised manuscript from Levesque S. et al. provides a more focused overview of a Prime Editing co-selection strategy based on ouabain treatment. The advantage over HDR and BE co-selection is well presented. Moreover, interesting observations about unexpected insertions/deletions after Prime Editing by PE3 and a strategy to reduce these unwanted genomic modifications is presented

I still have concerns about the points below:

Major:

1) I believe the presentation of results in Fig1 and 2 as merged data from the different experiments looks better than the original one. The figures show a clear trend of Prime Editing co-selection by Ouabain treatment. However, as pointed previously running experiments with $n \geq 3$ could facilitate a statistical analysis that is lacking throughout the manuscript. The lack of statistical analysis represents a weak point of this work.

2) The authors provide more details on the selection criteria for the Na⁺/K⁺ ATPase gain of function mutations that provide different level of resistance to ouabain. References to specific articles supporting these criteria are also provided. However, it is important to clarify that the provided references use random integration of over-expressing plasmids to evaluate Na⁺/K⁺ ATPase mutants. These experiments have little relevance in the context of precise modifications of endogenous genes. I believe that a functional analysis of selected clones/pools is required to support the claims that the mutant enzymes function normally. Examples of functional analysis are mRNA expression/Western Blot of the Na⁺/K⁺ ATPase and/or functional characterisation of the receptor activity.

3) The presence of insertions/deletions when using PE3 is an interesting observation and provides a new insight in Prime Editing mechanism. It also suggests DSBs repair pathway involvement in PE3. Considering this, translocations from the GOI to the Na⁺/K⁺ ATPase gene could arise as result of co-editing by PE3. To clarify this point, the authors could use a PCR/qPCR strategy with primers flanking predicted junction points on selected pools.

Minor:

4) The authors mentioned that to date six types of prime editors are used: PE2, PE3, PE3b, PE4, PE5 and PE5b. For completeness, I suggest adding references to Prime Editors that are exploiting Cas9 nucleases, see below:

Adikusuma, Fatwa, et al. "Optimized nickase-and nuclease-based prime editing in human and mouse cells." *Nucleic acids research* 49.18 (2021): 10785-10795.

Peterka, Martin, et al. "Harnessing DSB repair to promote efficient homology-dependent and-independent prime editing." *Nature Communications* 13.1 (2022): 1-9.

Tao, Rui, et al. "WT-PE: Prime editing with nuclease wild-type Cas9 enables versatile large-scale genome editing." *Signal transduction and targeted therapy* 7.1 (2022): 1-8.

5) As pointed out previously data presentation is often non-intuitive. An example is Figure 4 that provides only an anecdotal snapshot of selected insertions with little extra information. More details on these events could be obtained by long range sequencing on cell pools. Alternatively, I suggest excluding this Figure from the main article.

Reviewer #2:

Remarks to the Author:

In this revised manuscript the authors have significantly improved their manuscript. All my questions have been adequately addressed. In particular, the authors analyzed novel byproducts consisting of larger insertions and deletions in cell clones.

There are minor typos in the rebuttal.

Reviewer 2 point 1:

"We have previously investigated the impact of coselection on nuclease-driven off-targets and we did not detect overt off-target activity at previously characterized off-target sites (Agudelo et al, Nature Methods, 2027 PMID: 28417998). Finally, we cite to several papers showing..."

Point-by-point response to referees

*Please note that we have highlighted the changes made to the manuscript since the last submission using blue font for the text.

Reviewer #1:

We thank the reviewer for acknowledging the improvements made to our manuscript. We have performed additional experiments, as well as modifications to the text, to address their comments.

Major comments:

- I believe the presentation of results in Fig1 and 2 as merged data from the different experiments looks better than the original one. The figures show a clear trend of Prime Editing co-selection by Ouabain treatment. However, as pointed previously running experiments with $n \geq 3$ could facilitate a statistical analysis that is lacking throughout the manuscript. The lack of statistical analysis represents a weak point of this work.

We interpret that this remark relates to two issues; (i) formal statistical testing, (ii) number of independently performed experiments.

(i) It is our understanding that if n is small (e.g. $n=2$ or 3) it is better to plot the individual data points rather than showing error bars and statistics¹. In our case, the magnitude of the effect is large and unequivocal, as the reviewer acknowledges. In our opinion, the reader can better interpret our data with individual data points, as suggested^{1,2}. All datapoints for each figure/table are now provided in an Excel file; The “Source Data file”. We note that the information provided in the “Statistics for Biologists” collection in the Nature portfolio <https://www.nature.com/collections/qghhqm> supports our view.

1. Cumming, G., Fidler, F. & Vaux, D. L. Error bars in experimental biology. *J. Cell Biol.* **177**, 7–11 (2007).
2. Vaux, D. L. Know when your numbers are significant. *Nature* **492**, 180–181 (2012).

As pointed out by the reviewer, our dataset shows a clear trend of enrichment after ouabain coselection, as observed with 15 different pegRNAs targeting 7 loci of interest in 3 different cell lines. All these experiments were reproduced independently at different times with equivalent results, confirming that ouabain coselection robustly increases prime editing efficiencies. Hence, there is a solid basis that supports our observations and conclusions. We also note that the system has already been implemented successfully by independent researchers who have obtained our plasmids from Addgene and communicated their results to us.

Thus, we disagree with the statement that “The lack of statistical analysis represents a weak point of this work” and that this is a standard requirement in the field.

(ii) Throughout our manuscript, we have clearly reported the number of independently performed experiments ($n=2$ when targeting endogenous loci and $n=3$ when using the EBFP to EGFP reporter). Given the magnitude of the effects reported, in different cell lines with multiple pegRNAs, adding a third biological replicate would not only cost thousands of dollars but would further delay publication with little to no impact to the quality of our manuscript. Illustrative examples of pivotal studies in the genome editing field reporting data from biological replicates can be found below³⁻⁵. Of note, one article that the reviewer requested us to cite (Peterka, Martin, et al. "Harnessing DSB repair to promote efficient homology-dependent and-independent prime editing." *Nature Communications* 13.1 (2022): 1-9) uses a diphtheria toxin (DT)-based selection system that is conceptually like ours and reports some data in the primary and supplemental figures as the mean of $n = 2$ biologically independent replicates. We hope that the reviewer will agree that these papers, many others in the field, and ours, are scientifically sound and valid.

3. Chen, P. J. et al. Enhanced prime editing systems by manipulating cellular determinants of editing outcomes. *Cell* **184**, 5635–5652 (2021).
4. Kluesner, M. G. et al. CRISPR-Cas9 cytidine and adenosine base editing of splice-sites mediates highly-efficient disruption of proteins in primary and immortalized cells. *Nat. Commun.* **12**, 2437 (2021).

5. Shy, B. R. et al. High-yield genome engineering in primary cells using a hybrid ssDNA repair template and small-molecule cocktails. *Nat. Biotechnol.* (2022) doi:10.1038/s41587-022-01418-8.

- The authors provide more details on the selection criteria for the Na⁺/K⁺ ATPase gain of function mutations that provide different level of resistance to ouabain. References to specific articles supporting these criteria are also provided. However, it is important to clarify that the provided references use random integration of over-expressing plasmids to evaluate Na⁺/K⁺ ATPase mutants. These experiments have little relevance in the context of precise modifications of endogenous genes. I believe that a functional analysis of selected clones/pools is required to support the claims that the mutant enzymes function normally. Examples of functional analysis are mRNA expression/Western Blot of the Na⁺/K⁺ ATPase and/or functional characterisation of the receptor activity.

To address this issue, we performed additional experiments to assess cell fitness and ATP1A1 expression. First, we selected bulk populations of cells (pools) harboring either *ATP1A1*-Q118R, *ATP1A1*-T804N, or both *ATP1A1*-T804N/Q118R mutations (≥80% PE edited alleles in each pool). We monitored cell proliferation on these pools and observed no growth delay, confirming that the modifications introduced after successive rounds of prime editing do not affect cell fitness. In addition, we performed western blots on these cell pools and observed little to no decrease in ATP1A1 expression. These data have been included in the manuscript as Supplementary Fig. 7.

- The presence of insertions/deletions when using PE3 is an interesting observation and provides a new insight in Prime Editing mechanism. It also suggests DSBs repair pathway involvement in PE3. Considering this, translocations from the GOI to the Na⁺/K⁺ ATPase gene could arise as result of co-editing by PE3. To clarify this point, the authors could use a PCR/qPCR strategy with primers flanking predicted junction points on selected pools.

To test if co-editing via PE3 can drive the formation of translocations we used a previously-described nested PCR-based approach^{6,7}. We attempted to detect translocations between *ATP1A1* and two genes of interest (*EMX1* and *RUNX1*) used in our current study. As a positive control, we

swapped the pCMV-PE2 expression vector for pX330 (wild-type SpCas9). We could reproducibly detect diverse translocations between chromosome 1 (*ATPIA1* exons 4 or 17) and chromosome 2 (*EMX1*) and 21 (*RUNX1*) using wild-type SpCas9 but not with PE3 three days post-transfection. However, we could amplify and sequence rare events in PE3 samples from selected pools. We could directly sequence the amplicons via sanger sequencing further confirming the rarity of PE3-induced translocations. This decreased frequency of translocations corroborate the findings made when comparing nucleases vs paired-nickases strategies⁷. These data have been included in the manuscript as Supplementary Fig. 9 and sequences from the two unique translocation events detected by PE3 are provided in the Supplementary Material.

6. Piganeau, M. et al. Cancer translocations in human cells induced by zinc finger and TALE nucleases. *Genome Res.* **23**, 1182–1193 (2013).
7. Renouf, B., Piganeau, M., Ghezraoui, H., Jasin, M. & Brunet, E. Creating cancer translocations in human cells using Cas9 DSBs and nCas9 paired nicks. *Methods Enzymol.* **546**, 251–271 (2014).

Minor comments:

- The authors mentioned that to date six types of prime editors are used: PE2, PE3, PE3b, PE4, PE5 and PE5b. For completeness, I suggest adding references to Prime Editors that are exploiting Cas9 nucleases, see below.

We thank the reviewer for the suggestion and have added these references to the introduction of our manuscript.

As pointed out previously data presentation is often non-intuitive. An example is Figure 4 that provides only an anecdotal snapshot of selected insertions with little extra information. More details on these events could be obtained by long range sequencing on cell pools. Alternatively, I suggest excluding this Figure from the main article.

As the reviewer points out, our work “provides a new insight in Prime Editing mechanism”. It is by sequencing these large insertions present in clones that we were able to explain their formation

and demonstrate how to prevent them. As such, we believe that these data are important to include in the primary figures of our manuscript. Please note that we have provided their DNA sequences in the Supplemental Material for anyone to analyze. Each sequence is associated with a precise clone number and agarose gels provided as Supplementary Figures. In addition, this representation of data is reminiscent to the work of Chen et al.³ (See Figure 2).

3. Chen, P. J. et al. Enhanced prime editing systems by manipulating cellular determinants of editing outcomes. *Cell* **184**, 5635–5652 (2021).

Reviewer #2:

We thank the reviewer for its contribution to our manuscript and its description of our revised manuscript has ‘significantly improved’.

Minor Comment:

- There are minor typos in the rebuttal.

The typos have been corrected.